# A computational reward learning account of social media engagement

Björn Lindström 1✉, Martin Bellander 2, David T. Schultner[1], Allen Chang[3], Philippe N. Tobler[4] & David M. Amodio[1,5]

Social media has become a modern arena for human life, with billions of daily users worldwide. The intense popularity of social media is often attributed to a psychological need for social rewards (likes), portraying the online world as a Skinner Box for the modern human. Yet despite such portrayals, empirical evidence for social media engagement as reward-based behavior remains scant. Here, we apply a computational approach to directly test whether reward learning mechanisms contribute to social media behavior. We analyze over one million posts from over 4000 individuals on multiple social media platforms, using computational models based on reinforcement learning theory. Our results consistently show that human behavior on social media conforms qualitatively and quantitatively to the principles of reward learning. Specifically, social media users spaced their posts to maximize the average rate of accrued social rewards, in a manner subject to both the effort cost of posting and the opportunity cost of inaction. Results further reveal meaningful individual difference profiles in social reward learning on social media. Finally, an online experiment ($n = 176$), mimicking key aspects of social media, verifies that social rewards causally influence behavior as posited by our computational account. Together, these findings support a reward learning account of social media engagement and offer new insights into this emergent mode of modern human behavior.

[1] Department of Psychology, University of Amsterdam, Amsterdam, The Netherlands. [2] Center for Psychiatry Research, Department of Clinical Neuroscience, Karolinska Institutet, Stockholm, Sweden. [3] Department of Psychological and Brain Sciences, Boston University, Boston, MA, USA. [4] Zurich Center for Neuroeconomics, Department of Economics, University of Zürich, Zürich, Switzerland. [5] Department of Psychology, New York University, New York, NY, USA. ✉email: bjorn.r.lindstrom@gmail.com

What drives people to engage, sometimes obsessively, with others on social media? In 2019, more than four billion people spent[1] several hours per day, on average, on platforms such as Instagram, Facebook, Twitter, and other more specialized forums. This pattern of social media engagement has been likened to an addiction, in which people are driven to pursue positive online social feedback[2,3] to the detriment of direct social interaction and even basic needs like eating and sleeping[4,5].

Although a variety of motives might lead people to use social media[6], the popular portrayal of social media engagement as a Skinner Box for the modern human suggests it represents a form of reinforcement learning (RL)[7] driven by social rewards. Yet despite this common portrayal, empirical evidence for social media engagement as reward-based behavior has been elusive. In the present research, we developed and applied a computational approach to large scale online datasets of social media use to directly test whether, and how, reward learning mechanisms contribute to social media behavior. In doing so, we sought to provide new insights into this emergent mode of human interaction while testing a learning theory model of real-life human social behavior on an unprecedented scale.

In online social media platforms, feedback on one's behavior often comes in the form of a "like"—a signal of approval from another user regarding one's post[2]—which is assumed to function as a social reward. Indeed, several lines of research support the idea that "likes" engage similar motivational mechanisms as other, more basic, types of rewards such as food or money. In humans, brain imaging studies have consistently shown that likes[8,9] and other social rewards, are processed by neural and computational mechanisms closely overlapping with those processing non-social rewards[10–14]. Although neuroscientific studies are largely constrained to the laboratory, such findings suggest that social media use might reflect the process of reward maximization, similar to what is observed across species in response to non-social rewards.

Reward learning processes on social media platforms should also be evident in behavior. Indeed, the receipt of likes has behavioral consequences consistent with reward learning. For example, the number of likes received for a post predicts satisfaction with that post, and in turn, more self-reported happiness[15,16]. Similarly, a user's social media activity increases after a post, suggestive of reward anticipation[17], and users provide more social feedback to others after receiving feedback themselves[18]. In addition to its direct effect on reward, the subjective value of likes is also influenced by social comparison in a way similar to non-social rewards[3,19,20], suggesting that social rewards, just like non-social rewards[21], might be relative, rather than absolute in nature. Together, these existing studies support the idea that social media engagement reflects reward mechanisms.

However, as most studies of online social rewards to date utilize self-report methods[22,23], direct evidence for a social reward learning account of behavior on social media is lacking. Furthermore, studies that do apply RL approaches to social media data have typically not sought to delineate psychological mechanisms underlying social media use, but instead to optimize software that interacts with users (e.g., by training recommender systems[24]). In addition, results from the few studies that have taken a quantitative approach to human behavior are mixed. In one study, negative evaluation of a post—a type of social punishment—led to deterioration in the quality of future posts, rather than the improvement predicted by learning theory[25]. By contrast, in another study, receiving more replies for a post on a specific social media discussion forum predicted a subsequent increase in the time spent on that forum relative to others, consistent with learning theory[26]. Thus, it remains unclear whether basic mechanisms of reward learning can help explain actual behavior on social media.

In the present research, we directly test whether social media engagement can be formally characterized as a form of reward learning. By analyzing more than one million posts from over 4000 individual users on multiple distinct social media platforms (see Methods), we assess, using computational modeling, how the putative social rewards received for posts in the past (e.g., the likes received when posting a "selfie") can help explain future behavior. Our computational modeling approach allows us to explicitly test how cross-species reward learning mechanisms contribute to this uniquely human mode of social behavior[27]. We also confirm, using an online experiment resembling common social media platforms ($n = 176$), that social rewards causally influence behavior as predicted by our reward learning account.

Computational learning theory posits specific behavioral patterns that would characterize online behavior as an expression of reward learning. A seminal empirical insight is that when animals (e.g., rodents in a Skinner box) can select the timing of their instrumental responses (e.g., when and how often to press a lever), the latency of responding (the inverse of the response rate) is negatively related to the rate of accrued rewards[28]. That is, a lower reward rate produces longer response latencies. Reinforcement learning theory provides both a normative explanation and a mechanistic machinery for this regularity: the more reward one receives, the shorter the average latency between responses should be, because acting more slowly results in a longer delay to the next reward, and the cost of this delay—the opportunity cost of time—is directly related to the average reward rate[29]. As consequence, when animals have learned, through interaction with the environment, that the average reward rate is higher, actions should be made faster because further rewards would be foregone by slower and fewer responses.

Although this RL theory was developed to explain animal behavior in laboratory tasks, on timescales of seconds and minutes, the theoretical relationship between the average reward rate and response latency is not tied to a specific timescale. Consequently, if social media taps into basic learning mechanisms, social media behavior should exhibit the same relationship between response latency—the time between successive social media posts—and the (social) reward rate. In other words, we hypothesize that a type of real-life behavior, on timescales rarely, if ever, investigated in the laboratory, exhibits this key signature of reward learning. Finally, we confirm the causal influence of the social reward rate on posting response latencies with an online experiment, designed to mimic key aspects of social media platforms.

## Results

**Reward learning on social media.** We tested our hypothesis that online social behavior, in the form of posts, follows principles of reward learning theory in four independent social media datasets (see Methods) (total $N_{Obs} = 1,046,857$, $N_{Users} = 4,168$) with computational modeling. These datasets came from four distinct social media platforms, where people post pictures and, in response, receive social reward in the forms of "likes." In Study 1 ($N_{Users} = 2,039$), we tested our hypothesis in a large dataset of Instagram posts[30] (average number of posts per individual = 418, see Supplementary Table 1 for additional descriptive statistics). Instagram exemplifies modern social media, with over 1 billion registered users, and its format—focused primarily on simple postings and the receipt of likes as feedback—makes it a unique case study. However, because there are significant economic motives on Instagram and similar platforms[31], an assessment of

reward learning on Instagram could be limited to some extent by the possibility of fraudulent accounts and "fake likes," among other strategic uses[32]. We therefore replicated and extended Study 1 in Study 2 ($N_{Users} = 2,127$) with data from three different topic-focused social media sites (discussion forums focused on Men's fashion, Women's fashion, and Gardening, see Methods), where economic motives are less likely (average number of posts per individual = 91, see Supplementary Table 1 for descriptive statistics). Finally, we conducted an online experiment (Study 3, $N_{Participants} = 176$), designed to mimic key aspects of social media platforms, in which we manipulated the social reward rate to verify its causal impact on response latencies.

**Social rewards predict social media posting**. We conceptualized the act of posting on a social media platform (e.g., Instagram) as free-operant behavior in a Skinner box with one response option (e.g., a single lever), where responses are followed by reward (i.e., likes). As outlined, a key prediction from learning theory for such situations, in which the agent can decide when to respond, is that the latency between responses should be affected by the average rate of rewards[28,29]. Before formally testing our computational hypothesis, we evaluated, in two complementary and model-independent ways, whether social media behavior was sensitive to social rewards.

First, we drew inspiration from classic work in animal learning theory, which established that response rates, an aggregate measure of response latency, typically follow a saturating positive (i.e., hyperbolic) function of reward rates[28]. This relationship, known as the quantitative law of effect[28], is a signature of reward driven behavior (especially on interval schedules of reinforcement[28]). To directly test whether social media behavior exhibits this pattern, we compared how well a hyperbolic function explained the relationship between likes and response rates relative to a linear function (Supplementary Note 1). We found that the "quantitative law of effect" explained behavior better than a linear relationship in all four social media datasets (mean $R^2$: Study 1 = 0.43, Study 2: = 0.37, see Supplementary Note 1), demonstrating that an aggregate measure of response latencies on social media exhibits a classic signature of reward learning[28].

Second, we defined a high resolution measure of response latency ($\tau_{Post}$) as the time between two successive social media posts (similar to the interval between responses in human laboratory tasks and in animal free-operant behavior, see Fig. 1),

and tested whether $\tau_{Post}$ was predicted by the history of likes using Granger causality analysis (see Methods). Granger causality is established if a variable (e.g., likes) improves on the prediction of a second variable (e.g., $\tau_{Post}$) over and above earlier (lagged) values of the second variable in itself. To ascertain the selectivity of this method, we first applied it to simulated data from generative models where the ground truth was known (causality or no causality). We then fine-tuned the analysis parameters (the lag number, see Supplementary Note 2) to reliably detect Granger causality in data simulated from our reward learning model, which we introduce next, but not from models without learning (in which likes are unrelated to behavior, see Supplementary Note 2). Applying this optimized analysis method to the empirical data showed that likes Granger-caused $\tau_{Post}$ in all four datasets (Study 1: $\tilde{Z} = 23.65$, $p < 0001$; Study 2: Men's Fashion: $\tilde{Z} = 3.94$, $p < 0001$; Women's Fashion: $\tilde{Z} = 14.16$, $p < 0001$; Gardening: $\tilde{Z} = 6.78$, $p < 0001$). Together, these results demonstrate that the history of social rewards (i.e., likes) influenced both the rate and the time distribution of social media posting. Such reward sensitivity is a minimal criterion for more formally testing the explanatory power of learning theory.

**Modeling the dynamics of social media behavior**. Having established that social media behavior is sensitive to reward, we next developed a generative model, based on RL theory of free-operant behavior in non-human animals[29]. The key principle of this theory is that agents should balance the effort costs of responding and the opportunity costs of passivity (i.e., the posting-related rewards one misses while not posting) to maximize the average net (i.e., gains minus losses) reward rate[29]. The consequence is that average response latencies should be shorter when the average reward rate is higher. This prediction holds both when the amount of reward is a direct function of the number of responses (i.e., ratio schedules of reinforcement) and when rewards become available at specific time points (i.e., interval schedules of reinforcement).

Building directly on these principles, our $\bar{R}L$ model specifies how agents adjust the latency of their responses to maximize the average net reward rate $\bar{R}$ (see Methods and Supplementary Methods). Hence, the model provides a formal account of our hypothesis that behavior on social media conforms to basic principles of reward learning. Formally, the model conceptualizes social media use as a sequence of decisions regarding the latency

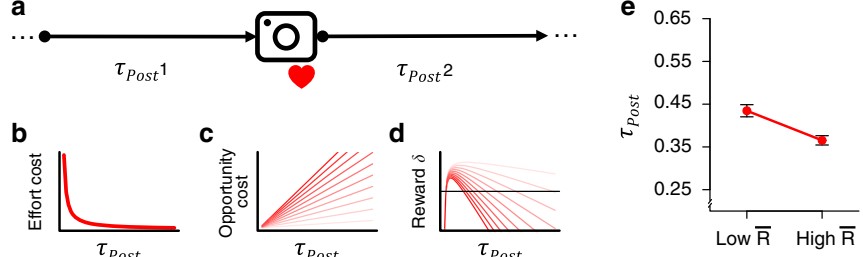

**Fig. 1 Schematic illustration of the computational hypothesis. a** The $\bar{R}L$ model describes how $\tau_{Post}$, the latency to the next social media post (denoted by the "camera" icon), is shaped by social rewards. Each post is followed by social reward (denoted by the "heart" symbol), which varies in number. The model adjusts the response policy, or threshold, which determines $\tau_{Post}$, to maximize the average net rate of reward. **b** The $\bar{R}L$ model posits an effort cost to responding (e.g., taking pictures, uploading), which decreases as a function of $\tau_{Post}$. The effort cost term penalizes posting in quick succession, because high effort reduces the average reward rate. **c** The opportunity cost of time increases as a function of the average reward rate $\bar{R}$. The gradient of red lines indicates increasing values of $\bar{R}$ (darker colors represent higher values), and thereby higher opportunity cost. **d** The optimal value of $\tau_{Post}$, which maximizes the net reward $\delta$, varies as function of $\bar{R}$ (darker colors represent higher values). The $\delta$ is used to update average reward rate $\bar{R}$. Note that the optimum, indicated by the peak of the function, moves to shorter response latencies when $\bar{R}$ is higher because the opportunity cost of time increases with $\bar{R}$. The horizontal line denotes 0. The figure assumes a constant effort cost $C$. **e** Simulated model predictions. The $\bar{R}L$ model predicts that $\tau_{Post}$, the latency between successive social media posts, will be shorter with high compared to low average reward rate, $\bar{R}$. The simulation involved $N = 1000$ independent synthetic individuals. The prediction is presented as estimated mean ± 99% CI from mixed-effects regression.

between successive posts, $\tau_{Post}$ (Fig. 1a)[29], where the agent maximizes the reward rate by adaptively adjusting $\tau_{Post}$ after observing each accrued reward. Psychologically, $\tau_{Post}$ can be thought of as the accumulation of motivation towards a threshold for posting (in similarity to the boundary in evidence accumulation models of decision-making[33]). The model policy, or threshold, which determines $\tau_{Post}$, is dynamically adjusted based on the net reward prediction error, $\delta$, the difference between the experienced reward and the reference level. The reference level is determined both by the individual's effort cost sensitivity (e.g., the subjective cost of taking pictures and uploading) and the subjective estimate of the average net reward rate $\bar{R}$[29,34], which determines the opportunity cost of slow responding (Fig. 1b, c). Both the effort cost and opportunity cost depend on the response latency, $\tau_{Post}$. In other words, the optimal response latency balances these two costs to maximize the net reward $\delta$ (Fig. 1d). The subjective estimate of $\bar{R}$ is updated using the same reward prediction error, thereby reflecting the integration of prediction errors across time[29]. In total, the model has three free parameters: learning rate, $a$; initial policy, $P$; and effort cost sensitivity, $C$ (see Methods). We verify with simulations that our $\bar{R}L$ model accurately reproduces standard patterns of animal behavior in Skinner boxes (Supplementary Methods and Supplementary Fig. 1). This demonstrates the validity of the $\bar{R}L$ model as an account of instrumental reward learning.

We simulated the model (~250,000 data points from 1000 simulated users, with random parameter values, see Supplementary Methods for details) to generate predictions for reward learning on social media. According to learning theory, $\tau_{Post}$ should be lower when the average reward rate is relatively higher. To verify this prediction in a simple manner, we rank-transformed and standardized $\bar{R}$ for each synthetic user and then dichotomized the variable at 0 to produce a qualitative "Low vs High $\bar{R}$" predictor (nearly identical results are observed with other definitions, see Supplementary Table 7). To facilitate subsequent comparison with empirical analyses, we summarized the simulated data using mixed-effects models. These analyses revealed a clear effect of low vs. high $\bar{R}$ on $\tau_{Post}$ ($\beta = 0.18$, SE = 0.007, $t = 31$, $p < 0001$), as expected. In other words, the model predicts (given the set of simulation parameters) that average response latencies should be ~18% longer when the average reward rate is low versus high (see Fig. 1e).

Our empirical analysis of the four social media platforms tested these model-based predictions with model estimation, statistical analyses, and generative model simulations. We optimized the parameters of the $\bar{R}L$ model for each individual user and quantitatively compared the explanatory value of the $\bar{R}L$ model to a null model without reward learning (see Supplementary Methods; model estimation and comparison procedure recovered the models with high probability, Supplementary Fig. 2). The null model assumes that posting on social media reflects a stable behavioral tendency (i.e., average response latency, one free parameter), which is not affected by reward. The model comparison provides a direct, quantitative test of reward learning as an explanation for social media use.

**Study 1**. We first modeled online behavior in the Instagram dataset of Study 1[30]. Model comparison showed that the $\bar{R}L$ model accounted better for the time distribution of responses ($\tau_{Post}$) than the model without learning for ~70% of the users (mean individual-level Akaike Information Criterion weight (AIC$_W$) = 0.7, 99% CI [0.68, 0.81], one-sample t-test relative to equal AIC$_W$ for the two models: $t(2038) = 23.1$, $p < 0001$, see Fig. 2a). The AIC$_W$ expresses the relative likelihood of one model over another[35]. Equivalently, Bayesian random effects model

comparison[36] showed that the $\bar{R}L$ model was more common than the model without learning (exceedance probability [xp] = 1), and classified ~70% of individuals as better explained by the $\bar{R}L$ model.

This conclusion was robust to the removal of individuals with especially short or long (outside the 20th and 80th deciles) average $\tau_{Post}$, or with few (or many) posts (see Supplementary Note 3), which confirms that the fit of the $\bar{R}L$ model was not driven by outliers. Similarly, splitting the dataset into four equally sized partitions showed that the $\bar{R}L$ model was the most common in all four partitions (mean AIC$_W$: 0.68–0.73, t-test against equal AIC$_W$: $t(508) = 9.63-13.9$, $p$s < 0001), which indicates that our conclusion is robust to sample idiosyncrasies and dataset size. Interestingly, we found that individuals with more Instagram followers exhibited non-linearly diminishing subjective value (utility) of likes, or in other words, derived less subjective value from each like (see Supplementary Note 4). This suggests that individuals with many followers might habituate to likes.

According to our theoretical framework, responses should be faster when the subjective reward rate is higher. Similar to how we derived model predictions (c.f., Fig. 1d), we used the model-based estimate of $\bar{R}$ (at $t-1$) dichotomized into "Low vs High" to predict the empirical $\tau_{Post}$ (at $t$), using log-linear mixed models (see Methods; the same conclusions are reached using continuous measures and regression models with cluster-corrected standard errors, see Supplementary Note 8 & Supplementary Table 7). In support of the hypothesis that people learn to maximize social rewards, the latency between posts, $\tau_{Post}$, was lower when $\bar{R}$ was relatively high (Instagram ($N_{Obs} = 851,946$, $N_{Users} = 2,039$): $\beta = -0.18$, SE = 0.003, $t = -54.59$, $p < 0001$, see Fig. 2b. Expressed in model comparison terms, the AIC$_W$ for the regression model including $\bar{R}$ was 1). Thus, increasing the average subjective reward rate from low to high reduced average latency between posts by ~18%, corresponding to ~8 h. Based on analysis with a continuous $\bar{R}$ term, this corresponds to a reduction of 0.34% (~5 min) in average posting latencies for each 1% increase in the subjective reward rate. Providing additional support for the logic of the $\bar{R}L$ model, the effect of $\bar{R}$ on posting latencies was stronger for individuals for whom the $\bar{R}L$ model provided a better fit (interaction Low vs High $\bar{R}$ * AIC$_W$ [centered at 0.5]: $\beta = -0.04$, SE = 0.008, $t = -5.5$, $p < 0001$). To ascertain the unique effects of our variables of interest, we adjusted for $R(t-1)$ (the number of likes received at post $t-1$), the specific post number, and the weekday of the preceding post in these analyses. We illustrate the relationship between $\tau_{Post}$, $\bar{R}$, and the model policy in Fig. 2c, d, which displays the posting behavior of an example individual over a period of two years. Together, these findings strongly indicate that online behavior conforms to principles of reward learning.

**Study 1: Alternative models**. To test the specificity of the $\bar{R}L$ model, we compared it with a set of plausible alternative models (see Supplementary Note 6). Specifically, we examined models (i) without effort cost ($C$) or net reward rate ($\bar{R}$) parameters, (ii) where the effort cost was fixed, or increased (rather than decreased) with post latencies, (iii) without an instrumental response policy, and (iv) based on foraging theory. Each of these provided worse accounts of the data than the $\bar{R}L$ model (see Supplementary Note 6, and Supplementary Tables 3–6).

**Study 1: model simulations**. In addition to demonstrating a model's superior fit to the data, strong support for a model depends on its ability to reproduce the effects of interest[37]. To confirm the model fitting results, we therefore generatively simulated the $\bar{R}L$ model (based on the median best fitting

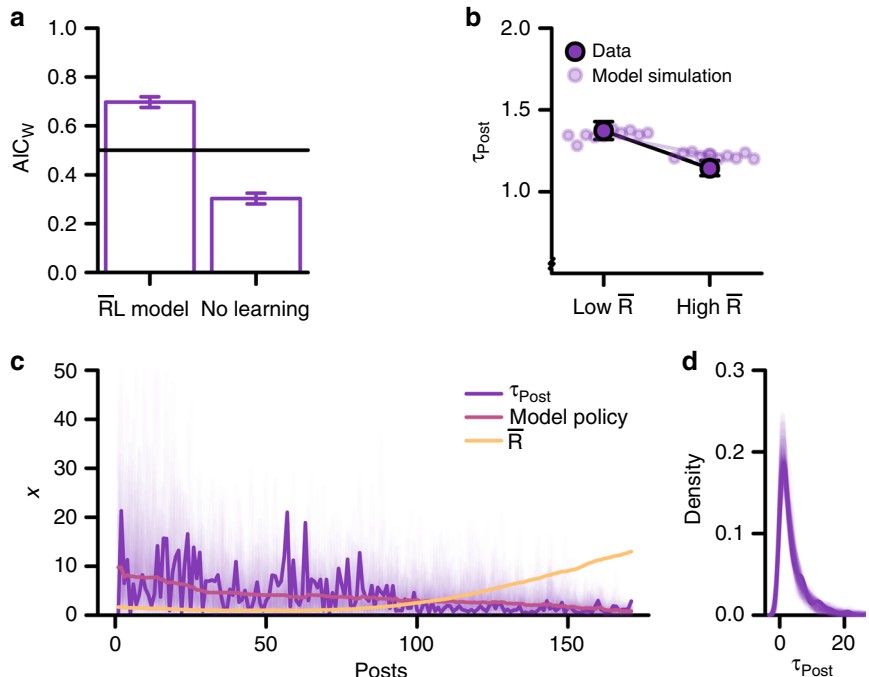

**Fig. 2 Behavior on Instagram is explained by reward learning (Study 1). a** Model comparison shows that the R̄L model explained behavior on Instagram ($N = 2,039$ independent individuals) better than a model without learning. The $AIC_W$ expresses the relative likelihood for each model, and are presented as means ± 99% CI. The horizontal line at 0.5 represents the chance level of no difference between models. The distribution of $AIC_W$ is displayed in Supplementary Fig. 3. Source data are provided as a Source Data file. **b** The model-derived estimate of R̄, the average reward rate, predicted the latency between posts ($N = 2,039$ independent individuals). As implicated by reward learning theory, the latency between posts was shorter with high compared to low R̄. Points indicate the corresponding estimates from synthetic data, based on ten generative simulation runs of the R̄L model (see text for details). The colored line denotes the average effect in the simulated data. Results are presented as means (fixed effects regression estimates) $+/-$ 99% CI from mixed-effects regression. **c, d** Model fit for an example individual. **c** The posting history of an individual user over 673 days was well approximated by the R̄L model. The model policy (or posting threshold) denotes the average response latency predicted by the model at a given time point. The faded purple lines show 100 simulations of $\tau_{Post}$ from the estimated model policy, which illustrate the expected degree of variability given that policy, and how the empirical $\tau_{Post}$ falls within this range. The yellow line indicates the model estimate of the net reward rate, R̄. Note that a higher estimated R̄ is associated with shorter response latencies ($\tau_{Post}$). See Supplementary Fig. 4 for additional example individuals. Source data are provided as a Source Data file. **d** The distribution of $\tau_{Post}$ for the same individual. The faded purple line shows 100 simulations of $\tau_{Post}$ from the estimated model policy. Source data are provided as a Source Data file.

parameters, but independent of the empirical data) and used mixed-effects models to summarize the simulations[38]. Notably, the simulation makes very limited assumptions of how likes were generated (i.e., as random draws from a Poisson distribution, with identical parameters for all individuals, see Supplementary Methods for additional details). Nonetheless, we found that the simple reward learning (R̄L) model reproduced the observed difference in response latency between high and low R̄ (Fig. 2b).

**Study 2.** To replicate and extend the results of Study 1, we collected data from three distinct social media sites (see Methods) which, in contrast to Instagram, focus on special interest topics (Men's fashion, Women's fashion, Gardening, respectively). Much activity on these social media sites is focused on textual exchange rather than images, but all three contain prolific "threads"—collections of posts focused on a specific topic—with predominantly image-based content (e.g., "What are you wearing today?", "Post pictures of your garden"), with many thousands of posts each. We limited our analyses to posts with user-generated images from such threads (see Methods and Supplementary Methods), but verify in the Supplementary Note 5 that the results are qualitatively identical when including text-based posts.

We again tested the hypothesis that social media behavior reflects social reward learning by estimating the same R̄L model as in Study 1. In all three datasets (190,721 data points from 2,127

individuals), regardless of the specific topic, model comparison favored the R̄L model over the model without learning (pooled mean $AIC_W = 0.77$, 99% CI [0.76, 0.79], t-test against equal model likelihoods: $t(2126) = 38.84$, $p < 0001$, $xp = 1$, see Fig. 3a–c). As in Study 1, we performed several robustness checks to verify this conclusion (see Supplementary Note 3). These findings converge with and generalize those of Study 1, providing platform-independent evidence that reward learning theory can help explain social media behavior.

As in Study 1, we used the model-based estimate of R̄ to predict the empirical $\tau_{Post}$, using mixed effects regression models (adjusting for the same covariates as in Study 1). As expected, a higher R̄ predicted faster responding in all three datasets (see Fig. 3d–f. Men's fashion ($N_{Obs} = 36,139$, $N_{Users} = 541$): $\beta = -0.08$, SE = 0.016, $t = -5.1$, $p < 0001$. Women's fashion ($N_{Obs} = 36,434$, $N_{Users} = 773$): $\beta = -0.16$, SE = 0.02, $t = -7.1$, $p < 0001$. Gardening: ($N_{Obs} = 118,148$, $N_{Users} = 813$): $\beta = -0.18$, SE = 0.02, $t = -12.09$, $p < 0001$. Thus, in these respective datasets, latencies between posts were 8%, 16%, and 18% shorter when the average reward rate was high rather than low. This corresponds, based on analysis with a continuous R̄ term, to a reduction of 0.18%, 0.41%, and 0.38% in average posting latencies for each 1% increase in the subjective reward rate. As in Study 1, the estimated effect of R̄ on posting latencies was stronger for individuals for whom the R̄L model provided a better fit (see Supplementary

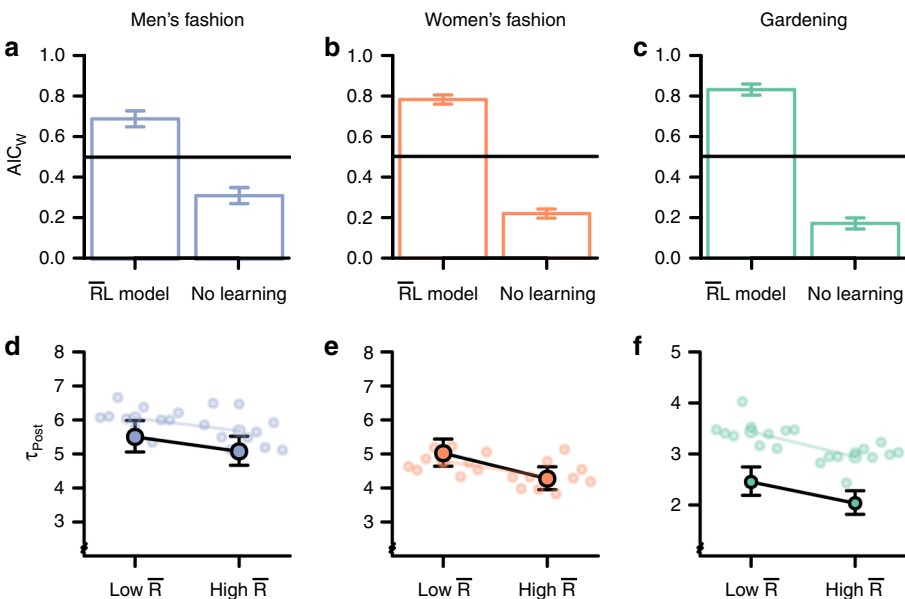

**Fig. 3 Signatures of reward learning on three social media sites (Study 2).** **a**–**c** Model comparison shows that the $\bar{R}$L model explained behavior on the three social media sites (total $N = 2{,}127$ independent individuals. **a** $N = 543$, **b** $N = 773$, **c** $N = 813$) better than a model without learning. The $AIC_W$ expresses the relative likelihood for each model, and are presented as means $+/-$ 99% CI. The horizontal line at 0.5 represents the chance level of no difference between models. The exceedance probability for the $\bar{R}$L model was 1 in all three datasets. The distribution of $AIC_W$ is displayed in Supplementary Fig. 3. Source data are provided as a Source Data file. **d**–**f** The model derived estimate of $\bar{R}$, the average reward rate, predicted the latency between posts on each social media platform (**d** $N = 543$, **e** $N = 773$, **f** $N = 813$ independent individuals). In line with reward learning theory, the latency between posts was shorter with high compared to low $\bar{R}$. The colored points indicate the corresponding estimates from simulated data, based on ten generative simulation runs of the $\bar{R}$L model (see text for details). The colored lines show the average effect in the simulated data. Results are presented as means (fixed effects regression estimates) ± 99% CI from mixed-effects regressions.

Note 7). Thus, regardless of platform topic, social media behavior conforms to model-based principles of reward maximization through RL.

We again conducted generative model simulations of the $\bar{R}$L model, based on the median estimated parameters, to test whether the model reproduced the effect of average reward rates on posting latencies. Corroborating Study 1, these simulations showed that the $\bar{R}$L model accurately reproduced the effect of $\bar{R}$ observed in the data (Fig. 3d–f) (but note that the absolute level of $\tau_{Post}$ is slightly overestimated in the Gardening dataset). Together with Study 1, these results confirm that basic reward learning theory provides a powerful tool for predicting and explaining the dynamics of social media use, independent of topic.

**Study 2: Social comparison in social reward learning**. The preceding analyses showed that people dynamically adjust their social media behavior in response to their own social rewards, as predicted by reward learning theory—a theory originally developed to test the effects of nonsocial rewards (e.g., food) in solitary contexts. However, given the intrinsically social context of social media use, we speculated that reward learning online could be modulated by the rewards others receive. In the Supplementary Note 9 we provide preliminary support for the hypothesis that reward learning, at least on the social media platforms we analyzed in Study 2, may be modulated by social comparison.

**Individual differences in reward learning on social media.** Having established that reward learning can help explain social media behavior, we next asked whether individuals differ in the ways they learn from rewards on social media. To address this issue, we used the parameter estimates of the basic $\bar{R}$L model as a compact but rich description of the mechanisms underlying behavior—a kind of computational phenotype (which is

behavioral in nature and makes no direct reference to the underlying genotype)[39]. Individual differences in these parameters can thus be viewed as differences in computational mechanisms[39] that are interpretable across domains. For example, individual differences in learning rates have previously been linked to both genetic[40] and developmental differences[41] between individuals, while individual differences in effort cost sensitivity have been related to the dopaminergic system[42].

More specifically, we used the three parameters of the original $\bar{R}$L model estimated for each individual from Study 1 & 2 (total $N_{Users} = 4{,}168$), as input for k-means clustering, an unsupervised, data-driven method for finding sub-groups in multidimensional data. Quantitative assessment, using multiple standard criteria, showed that four clusters gave the best sub-group solution (see Fig. 4a and Methods). In Supplementary Note 10, we report additional robustness analyses, which show that this cluster solution was stable. These clusters comprised between 41% (1739 individuals) to 7% (299 individuals) of the total dataset. Importantly, although the four datasets varied in mean $\tau_{Post}$ (as reflected in the $P$ parameter), the cluster assignment was not strongly explained by dataset (Cramér's $V = 0.3$; Cramér's $V$ is a measure of the association between two nominal variables, where 1 denotes perfect association). This indicates that clusters captured individual differences in computational learning mechanisms, rather than idiosyncrasies of social media sites.

Figure 4b illustrates the four putative computational phenotypes. For example, individuals in cluster 1 are characterized by a relatively low learning rate ($a$). Such individuals are especially insensitive to social rewards in their behavior (and naturally, the $\bar{R}$L learning model provided the worst fit to these individuals relative to the model without learning: mean $AIC_W = 0.11$, vs $AIC_W \sim 0.77$ in the other three clusters). By comparison, individuals in cluster 2 are characterized by low effort cost and

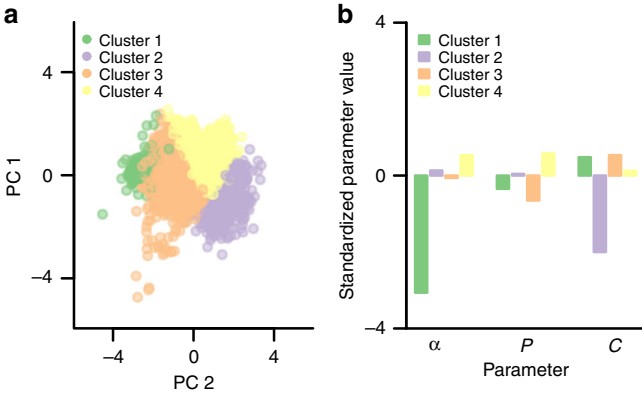

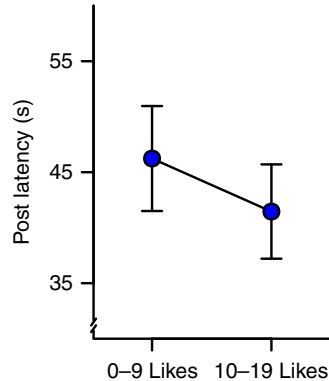

**Fig. 4 Computational phenotypes in reward learning on social media. a** Cluster analysis (n = 4168 independent individuals from Study 1 & 2) of the estimated R̄L model parameters indicated that there were four distinct individual difference profiles in reward learning on social media. For illustration, the cluster assignments are plotted on the two first principal components (PC). Source data are provided as a Source Data file. **b** Profile of median R̄L model parameter values (standardized) for each cluster; $a$ = learning rate, $P$ = initial response policy, $C$ = effort cost sensitivity. Source data are provided as a Source Data file.

**Fig. 5 Experimental manipulation of social reward rates (Study 3).** The estimated effect of social reward rate condition on posting latencies in the online experiment (n = 176 independent individuals, see Supplementary Fig. 5 for a design overview). Results are presented as means (fixed effects regression estimates) ±1 SE of the regression estimates from from mixed-effects regression.

average learning rate, whereas cluster 4 exhibits the opposite relationship between learning rate and effort cost (and cluster 3 an intermediate profile). Individuals in both clusters 2 and 4 therefore readily post in response to social rewards, although the underlying mechanisms differ. In summary, the computational phenotyping indicates that there are important individual differences in the mechanisms underlying social media behavior.

**Study 3**. Finally, to provide direct evidence that the social reward rate affects posting latencies, we conducted an online experiment in which we experimentally manipulated social rewards and observed posting response latencies. The experiment was designed to capture key aspects of social media, such as Instagram. Participants (n = 176) could post "memes"—typically, an amusing image paired with a phrase that is popular on real social media—as often and whenever they wanted during a 25 min. online session (total number of posts = 2,206, see Supplementary Methods for details). Participants received feedback on their posts in the form of likes (0–19) from other ostensible online participants ("users", see Supplementary Fig. 5 for an overview of the experiment). Participants themselves could also indicate "likes" for memes posted by other users. To test whether a higher social reward rate causes shorter response latencies in posting, we increased or decreased the average number of likes participants received between the first and second halves of the session (low reward: 0–9 likes/post, high reward: 10–19 likes/post, drawn from a uniform distribution, with direction of change counter-balanced across subjects). As expected, mixed effects regression (see Methods) showed that the average post latency was longer when the social reward rate was lower (0–9 likes/post) relative to higher (10–19 likes/post): $\beta = 0.109$, SE = 0.044, $z = 2.47$, $p = .013$ (see Fig. 5), corresponding to a 10.9% difference. Notably, participants who reported more followers on Instagram exhibited weaker effects of likes on their behavior (see Supplementary Note 11). This finding parallel how individuals with more Instagram followers in Study 1 exhibited more diminished marginal utility of likes (see "Study 1" above and Supplementary Note 4). These results further support the validity of our experiment in assessing the psychology of real-world social media use. We report additional analyses and robustness checks in Supplementary Note 11.

To directly relate the experimental results of Study 3 to our model-based analyses of online behavior in Studies 1 and 2, we used the R̄L model to generate subjective R̄ time series for the subset of participants with a sufficient number of responses (see Supplementary Note 11 for details), and used these (instead of reward condition) to predict response latencies. In accordance with the model fits to the real social media data (Study 1-2), the average response latency was longer when the subjective reward rate was low, relative to high (mixed effects regression, $n = 156$: $\beta = 0.28$, SE = 0.045, $z = 6.24$, $p < 0001$). These experimental results demonstrate that social rewards causally influence response latencies, in support of our conclusion that social reward rates shape real social media behavior.

**Discussion**

In an age where our social interactions are increasingly conducted online, we asked what drives people to engage in social media behavior. Across two studies of four large online datasets, we found that social media behavior exhibited a signature pattern of reward learning, such that computational models inspired by RL theory, originally developed to explain the behavior of non-human animals, could quantitatively account for online behavior. This account was further supported by experimental data, in which manipulated reward rates affected the latency of social media posted in line with this reward learning model. Together, these results provide an important advance in our understanding of people's use of online social media, an increasingly pervasive and profoundly consequential arena for human interaction in the 21st century.

Our results provide clear evidence that behavior on social media indeed follows principles of reward maximization, and thereby give credence to the popular portrayal of social media engagement as a Skinner Box for the modern human. These observations, along with their formal modeling, have broad implications for understanding and predicting multiple aspects of online behavior, including dating (e.g., learning from outcomes on dating apps), social norms, and prejudice[43]. For example, it has been argued that online expressions of moral outrage, and in turn polarization[44], are fueled by social feedback, such as likes, in accordance with the principles of reward learning[45]. Our findings and theoretical framework provides a plausible mechanistic basis for such processes, and thereby further expand the scope of simple reinforcement learning mechanisms for explaining seemingly complex social behaviors, such as social exclusion[38], behavioral traditions[46], and socio-cultural learning[47].

Our computational model of social media behavior draws from RL theory originally intended to explain how non-human animals select the vigor of their responses by encoding the average net rate of rewards[29]. Apart from providing a normative explanation for key behavioral regularities (e.g., the "matching law"[28]), an important aspect of this theory is the idea that $\bar{R}$, the average rate of reward, is encoded by the tonic, average level of dopamine[29]. This idea has received some support in humans, where pharmacologically increasing the tonic level of dopamine, which according to the theory corresponds to a higher subjective reward rate, decreased average response latencies[48]. Although our behavioral findings cannot speak to the neurobiological basis of reward learning on social media, the link we establish between online response latencies and the average social reward rate warrant further exploration of the underlying brain mechanisms[14].

More generally, our results indicate that dopamine inspired RL theory may help to explain real-life individual behavior on timescales that are orders of magnitude larger than typically investigated in the lab. In turn, this insight might contribute to a more mechanistic perspective on both healthy and maladaptive (e.g., addictive[4,5]) aspects of social media use, with the potential to inspire novel, theoretically grounded design solutions or interventions. Such interventions could be individualized by applying computational phenotyping to an individual's existing social media record (e.g., by increasing the effort cost of posting for individuals characterized by low $C$), thus providing ideographic approaches developed from theoretical models tested on large-scale data. Although our data do not speak directly to whether intense social media use is maladaptive or addictive in nature, it suggests a new approach for asking such critical questions.

Naturally, there are many possible reasons for posting on social media in addition to reward seeking, ranging from self-expression to relational development[6]. While our research focused on how social rewards explain behavior, it does not preclude the potentially important roles of other motivations. Incorporating relational considerations in the $\bar{R}$L model, such as reciprocity or network proximity, represents an important goal for future research. Nevertheless, the learning model tested here explained behavior well, suggesting that reward learning is a major factor in social media engagement.

Our results raise several new questions regarding the role of reward in social media behavior. First, while our analysis of anonymous, real-world social media data precluded demographic characterization of users, it is possible that certain demographic factors, such as age, may moderate the effect of reward learning in online behavior. For example, adolescents tend to be more sensitive than adults to social rewards and punishments[49], and thus our results may be particularly informative to questions of adolescent social media behavior. An examination of the developmental trajectories of the computational phenotypes in social reward learning identified here, and their relations to individual differences in psychological traits, could further illuminate age effects in online behavior. Furthermore, while our research focused on the effect of social rewards (i.e., likes) on posting behavior, negative feedback, which is rampant on many social media platforms (e.g., down votes), is also likely to drive learning. The RL framework we proposed here may also be extended to include such social punishments. For example, treating social punishments as reinforcement with negative utility would, in principle, allow direct application of the $\bar{R}$L model in its current form, but it is possible that additional motivational factors, such as negative reciprocity, also play an important role in aversely motivated social media behavior. In addition, as we focused on the timing, rather than the content (e.g., of images or comments), of social media posts, an important future goal will be to characterize how people learn to produce actions (e.g., posting

content, comments on others' posts) that maximize reward. Our $\bar{R}$L model could be modified to include action selection as a part of the response policy[29]. Finally, our analysis in Study 2 suggests that social comparison may contribute to reward learning on social media by providing a social reference level for the number of likes required to elicit a positive reward prediction error. Although this result comports with prior research examining social comparison on social media[3,23], as well as neural reward processing[19], the correlational nature of the big data used here necessitates caution, as other explanations cannot be ruled out. These findings present an opportunity for future experimental research to establish the causal nature of social comparison and to explore its boundary conditions.

In conclusion, our findings reveal that basic reward learning mechanisms contribute to human behavior on social media. Understanding modern online behavior as an expression of social reward learning mechanisms offers a new window into the psychological and computational mechanisms that drive people to use social media while illuminating the link between basic, cross-species mechanisms and uniquely human modes of social interaction.

## Methods

**Social media datasets**. Study 1 was based on data from a previously published study (see[30] for further information), in which data collection was based on a random sample of individuals who partook in a specific photography contest on Instagram in 2014. We find no evidence that contest participation was related to posting behavior (see Supplementary Note 12). The dataset was fully anonymized. To allow for analyses of learning, we excluded all individuals with less than 10 posts[50]. For Study 1, the final dataset consisted of 851,946 posts from 2,039 individuals. For Study 2, we obtained three datasets from three different topic-focused (Men's fashion: *styleforum.net*, Women's fashion: *forum.purseblog.com*, Gardening: *garden.org*, see Supplementary Methods for details) social media discussion forums using web scraping techniques[51] on publicly accessible data. The datasets were fully anonymized, and only included the time stamps and likes associated with posts in prolific threads focused on user-generated images (e.g., pictures of one's clothing, see Supplementary Methods). For our analyses, we focused on posts with user-generated images, and, in analogy to Study 1, removed all users with fewer than ten image posts. The Study 2 dataset consisted of 190,721 posts from 2,127 individuals (Men's fashion: $N = 543$, Women's fashion: $N = 773$, Gardening: $N = 813$). This research was conducted in compliance with the US Office for Human Research Protections regulations (45 CFR 46.101 (b)).

**Experiment**. The experiment was conducted on Amazon Mechanical Turk, and approved by the ethical review board of the University of Amsterdam, The Netherlands. See the Supplementary Methods for additional information.

**Description of the $\bar{R}$L model**. The $\bar{R}$L model is a policy gradient version of R-learning[52]. Rather than storing action values for options and using these as a basis for decision-making, the $\bar{R}$L model directly updates a parametrized response policy (the mean parameter of an exponential distribution). This is beneficial for learning problems with continuous action spaces (e.g., the latency between responses)[53]. In close similarity to standard RL models for discrete action spaces, the $\bar{R}$L model incrementally learns to adjust its actions (i.e., $\tau_{Post}$) from prediction errors. In contrast to standard RL approaches in psychology, the prediction errors are used to directly adjust the response policy to maximize the undiscounted net rate of reward rather than to update action values.

For each post, the $\bar{R}$L model selects $\tau_{Post}$ as a draw from an exponential distribution, where the mean (i.e., the response policy or threshold) is dynamic:

$$\tau_{Post^t} = e^{\text{Policy}^t - \alpha * \bar{R}^t} \tag{1}$$

The initial response policy (i.e., for $t = 1$) was estimated as a free parameter ($0 \leq P \leq \infty$). The subtraction term in Eq. (1) implements the momentous effect of the average reward rate (e.g., changes in motivational state), $\bar{R}^t$, on the response rate, which is independent of learning[29]. This term, which can be thought of as "Pavlovian" (since it is independent of instrumental behavior), slightly amplifies the effect of the average reward rate on $\tau_{Post}$. Model comparison and simulation showed that this term has a small but significant contribution to model fit, but does not affect qualitative predictions.

The response policy, which determines $\tau_{Post}$ is adjusted based on the prediction error, $\delta$, the difference between the experienced reward ($R^t$) and the reference level:

$$\delta^t = R^t - \frac{C}{\tau_{Post^t}} - \bar{R}^t * \tau_{Post^t} \tag{2}$$

The reference level explicitly takes into account both the effort cost of fast responding and the opportunity cost of slow responding (Fig. 1b, c), which is

determined by the subjective estimate of the average net reward $\bar{R}$[29,34]. In our application of the model, $C$ $(0 < C \leq \infty)$ is a user-specific parameter that determines the subjective effort cost of posting (e.g., taking pictures, uploading). This parameter penalizes posting in quick succession (e.g., posting three times in one day is more costly than posting three times in three days; we evaluate different effort cost formulations in the Supplementary Note 5). If actions have no intrinsic cost, the optimal policy would be to respond as quickly as possible[29,54]. The last term in Eq. (2) characterizes the opportunity cost of slow responding, which increases with $\bar{R}$ (Fig. 1c).

The $\bar{R}$L model seeks to maximize reward by dynamically updating the response policy by the "gradient", or slope, of reward. In short, the gradient indicates the direction and distance to the hypothetical maximum net reward that could have been accrued at time $t$. To compute the gradient, the model tracks the sequential difference between responses (i.e., draws from the policy, which can be thought of as exploration), and combines this quantity with the net reward prediction error (Eq. 2):

$$\Delta \tau_{Post^t} = \tau_{Post^t} - \tau_{Post^{t-1}} \quad (3)$$

$$Policy^{t+1} = Policy^t + \alpha * \Delta \tau_{Post^t} * \delta^t \quad (4)$$

Thereby, the model can learn the instrumental value of slower vs. faster responses. For example, if $\Delta \tau_{Post^t}$ is positive, which would be the case if the response latency at $t$ was longer than at $t-1$, but $\delta^t$ negative, which indicates that the net reward was lower than on average, the gradient (Eq. 4) is negative. This results in a reduced response policy (the degree of adjustment depends on the magnitude of both $\Delta \tau_{Post^t}$ and $\delta^t$), and in turn shorter future response latencies. In contrast, if both $\Delta \tau_{Post^t}$ and $\delta^t$ are either positive or negative, the response policy will increase, which leads to longer response latencies. The average reward rate is updated using the same reward prediction error as the policy, as it directly reflects net reward value[52]:

$$\bar{R}^{t+1} = \bar{R}^t + \alpha * \delta^t \quad (5)$$

However, regardless of whether slower or faster responses are rewarded, an increase in the average reward rate (Eq. 5) results in a higher opportunity cost of time (Eq. 2), which in turn results in shorter response latencies (Eq. 2, and Fig. 1d).

If either $\Delta \tau_{Post^t}$ or $\delta^t = 0$, the model can theoretically be trapped in local reward minima. However, the stochastic policy (Eq. 1) ensures that $\Delta \tau_{Post^t}$ is different from 0 with exceedingly high probability, which promotes continuous search by sacrificing convergence if the step size parameter is fixed (as the model policy will continuously change). For simplicity, the same step size parameter $a$ $(0 < a \leq 1)$ was used for all update terms (Eq. 1 & 4–5). Inclusion of separate step size parameters for the different update equations did not reliably improve (parameter number penalized) model fit. Additional model information and estimation methods are detailed in the Supplementary Methods.

**Statistical analysis**. All model estimation, simulations, and statistical analyses were conducted using R. All reported p-values are two-tailed. Granger causality analysis was applied to first differenced data using the *plm* package for panel-data analysis (see Supplementary Methods for details)[55]. Mixed effects modeling was conducted with the *lme4*[56] and *glmmTMB*[57] packages. All log-linear mixed effects models included a random intercept for each user. In the statistical analyses, the dependent variable $\tau_{Post}$ was log transformed (as the time between events follows an exponential distribution) to improve linearity. All predictors were standardized within individual (i.e., centering within cluster) to produce individual-level estimates[58]. Degrees of freedom, test statistics, and p-values were derived from Satterthwaite approximations in the *lmerTest* package[59]. The key statistical analyses were in addition repeated using log-linear regression models with cluster-corrected standard errors to ensure robustness (see Supplementary Table 7). Prior to k-means clustering, the $\bar{R}$L model parameter estimates were log-transformed (to improve linearity) and standardized. The optimal number of clusters was determined using the *NbClst* package[60].

**Reporting summary**. Further information on research design is available in the Nature Research Reporting Summary linked to this article.

## Data availability
The availability of the dataset analyzed in Study 1 is described in ref. [30]. The datasets analyzed in Study 2 are available on reasonable request from the corresponding author. The dataset analyzed in Study 3 is available on the Open Science Framework (https://osf. io/765py/). A reporting summary for this article is available as a Supplementary Information file. Source data are provided with this paper.

## Code availability
Code for estimating the $\bar{R}$L and No Learning models is available at the Open Science Framework (osf.io/765py/).

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

## Acknowledgements

We thank Andreas Olsson for helpful comments on an earlier version of the manuscript, and Lucas Molleman for valuable suggestions concerning the experimental design. Work on this article was supported by grant from the Netherlands Organization for Scientific Research to DMA (VICI 016.185.058). PNT acknowledges funding support from the Swiss National Science Foundation (100019_176016).

## Author contributions

B.L. conceived of the study. B.L. developed the study in discussion with M.B., P.T. and D.A., B.L. designed and implemented the computational models, and conducted all analyses. M.B. and A.C. collected and processed the data for Study 2. B.L., D.S. and D.A. designed the behavioral experiment. D.S. coded the behavioral experiment and collected data. B.L. and D.A. wrote the manuscript. B.L., P.T. and D.A. edited the manuscript and contributed to the interpretation of the results.

## Competing interests

The authors declare no competing interests.
