## [Peer Review File · Nature Communications]

Reviewers' comments:

Reviewer #1 (Remarks to the Author):

Lindstrom and colleagues test whether reinforcement learning mechanisms contribute to behavior on social media. Across a large sample of individuals they apply computational models of reinforcement learning and suggest that such behaviour on social media conforms to principles of reward learning. They also find individual differences in learning are related to a users tendency for social comparison. This is a well written and well-conducted study that aims to answer an important question. I enjoyed reading it and I applaud the authors attempt to explain social media behavior computationally. However, I have some major concerns as to whether the study shows specificity to reinforcement learning mechanisms and the lack of any experimental manipulation.

Major

Specificity to RL mechanisms – the authors focus their model comparison on two models, an RL model and a model with no learning. However, they do not falsify whether they have simply assessed a limited model space (e.g. Palminteri et al., 2017). It could just be they haven't compared relevant models. Of course this can become a bit intractable as there are potentially unlimited models of behaviour, but there are other plausible models that come to mind. For example, theories of optimal foraging (Charnov, 1976) describe how make foraging decisions are based on the reward rate, the richness of the environment (the average reward rate) and the dispersion of patches. Can the authors show that foraging models, that do not contain a learning component, account for the behavior? I appreciate they ran simulations to show parameter recovery but since the main aim of the paper is to show that social media behavior conforms to RL principles it is very important to show specificity.

Machine learning approaches to understand behaviour on the internet are not necessarily new and are already used by many companies to evaluate how to get people to buy products online and personal ad recommendations (e.g. Theocharous et al., 2015, Personalized Ad Recommendation Systems for Life-Time Value Optimization with Guarantees; Kalyanam J, Quezada M, Poblete B, Lanckriet G (2016) Prediction and Characterization of High-Activity Events in Social Media Triggered by Real-World News. PLOS ONE 11(12): e0166694. <https://doi.org/10.1371/journal.pone.0166694>). These papers compare several different models with different parameters. In general, there seems to be a relatively large field of machine learning approaches to social media behaviour that is omitted in the manuscript where lots of different types of sophisticated models beyond simple RL that are used to predict internet behaviour.

Theoretically, I am not sure RL principles to understand behavior on Instagram make much sense. Simple

RL models describe how associations between stimuli/actions and outcomes are paired over time, but in the authors' model they instead use the average reward rate? Or I may have misunderstood something here.

I feel that without an experimental manipulation of the participants in the study the conclusions drawn about how social media behavior conforms to RL principles is too much of a leap. The authors could conduct an online study of the participants whose posts they collected and show, for example, that those more sensitive to reward in an independent RL task are those who show more reward sensitive behavior on social media. Or they could set up some kind of simulated Instagram environment where they can control the reward rate and measure the influence on people's behaviour. I appreciate this is difficult to do but without it the conclusions of the paper are a bit strong I feel.

Minor

The authors should tone down some of the bolder claims in the manuscript as they did not experimentally manipulate any behavior. For example sentences such as 'Together, our results expand the explanatory reach of learning theory from the behavior of rodents in lab experiments to the behavior of humans on social media'. And 'the RL model as a compact but rich description (i.e., a computational phenotype) of the mechanisms underlying behavior³⁸.' There is not any evidence that peoples genotypes are interacting with the environment in the paper, which would be necessary for a computational phenotype.

Reviewer #2 (Remarks to the Author):

In their manuscript, Lindström et al. used computational modeling to examine whether behavior on social media follows classic reward reinforcement learning (RL) patterns. Anonymous data was collected from 4 social media platforms across two studies (Instagram and three blog forums) and was submitted to various learning models. Briefly, the authors found that posting behavior (rate of posting and latency between posts) across the social media platforms was best explained by RL models with parameters that accounted for social rewards ('likes' and comments) on the posts, the effort costs of increased action, the opportunity costs of reduced action, and reward prediction error, consistent with animal models of reward learning. Furthermore, they find that social comparison (e.g., number of 'likes' on one's own post relative to recent posts by others) can explain some of the individual differences in reward learning.

This is a timely and well-written manuscript that applies computational affective methodology to ecologically salient behaviors in a uniquely human social setting. The authors convincingly argue their thesis by demonstrating clear and considerate comparisons with other models and they nicely address other possible confounding variables with additional analyses. Overall, the manuscript puts forth novel questions and findings that will be of interest and pertinent across multiple scientific disciplines with

important implications that reach beyond academia to a range of industries in society. There are some issues and clarifications that could augment the impact of the paper and are listed below.

- The introduction touches on recent assertions that social media engagement follows behavioral patterns that can be compared to addiction. It is important for the authors to qualify in their discussion the ways in which their results not only do not directly provide support for this claim, but also that there are other possible confounding variables at play and other interpretations of the results. For example, the nature of the feedback in these datasets (other than dichotomous categorization of 'likes' vs comments) is unknown, including the valence (positive/negative) and content (affective/informative, particularly in the forums) of the comments. This is especially critical as negative feedback (a social punishment) may deter social media engagement as referenced in Cheng et al., 2014. As another example, social rewards may or may not motivate other social media behaviors (e.g., viewing others' posts, 'liking' and commenting on others' posts), which are not accounted for in this study. Furthermore, it is unknown based on these results whether RL-based behaviors on social media have adaptive or maladaptive physical and psychological outcomes, which would speak more directly to the external claims that social media behavior follows patterns similar to addiction.

- Another important confounding variable that should be discussed is age of the users. Instagram, for example, is wildly popular among adolescents (ages 13-17) and emerging adults (ages 18-24). Relatedly, these populations have also been shown to be more sensitive to social rewards³. Do the authors have access to this information in any dataset? It would be interesting information to add if available. It would behoove the authors to discuss the possible effects of user age as it relates to their findings and broader implications of this work.

- It would be beneficial to the reader if the authors were to report descriptive statistics for their subjects and primary variables of interest (e.g., how many followers the users have, how often they post, how much feedback was given across the different platforms).

- The RL model had worse fit for individuals with many followers on Instagram. The authors posit that individuals with larger followings may be motivated by monetary rewards, and these rewards often only come into play when individuals have followings between 5,000 and 100,000+ by some estimates^{4,5} (hence one reason why it'd be helpful to have descriptive statistics on these users for comparison). Another possible interpretation of these findings is that individuals with larger followings, on average, receive consistently higher amounts social rewards (e.g., 'likes'), and have therefore become habituated to the feedback.

- That the data collected on Instagram originated from individuals who partook in a contest should also be further discussed. It is unclear whether these individuals partook in this and other contests often, and how their participation in these online contests might affect their posting behavior.

- It would be useful for the authors to comment further on the implications of the 4 computational phenotypes. What might attribute to such individual differences? What might this suggest about social

media engagement on a broader level?

- Regarding the social comparison model, it would be helpful to clarify how the current model differs from models such as the Fehr-Schmidt inequity aversion model. Also, how were disadvantageous and advantageous inequality assigned (based on post content, etc?)

1. <https://www.pewinternet.org/2018/05/31/teens-social-media-technology-2018/>
2. <https://www.pewresearch.org/fact-tank/2019/04/10/share-of-u-s-adults-using-social-media-including-facebook-is-mostly-unchanged-since-2018/>
3. Foulkes, L., & Blakemore, S. J. (2016). Is there heightened sensitivity to social reward in adolescence? *Current opinion in neurobiology*, 40, 81-85.
4. https://www.huffpost.com/entry/make-money-on-instagram_n_55ad3ad6e4b0caf721b3624c
5. <https://www.forbes.com/sites/forbescoachescouncil/2017/11/20/how-to-make-money-off-your-instagram-account/#4f7e88fc6af0>

Reviewer #3 (Remarks to the Author):

Lindstrom and colleagues present data testing a hypothesis that reinforcement learning mechanisms contribute to the social media behavior. Using data from Instagram and online forums, they show that the frequency of posting is related to the number of “likes” received for posts. They fit a variety of reinforcement learning models to the data and evaluate whether there are distinct “computational phenotypes” occupying different regions of parameter space in the RL models. They further consider how the distribution of computational phenotypes can be explained by individual differences in sensitivity to social comparison. The authors conclude their findings “offer new insights into this emergent mode of human behavior on an unprecedented scale”.

There is a lot to like about this paper. Fitting computational models to behavior on social media is novel and interesting. I appreciated the authors’ efforts to link their approach with animal models, and the replications across multiple social media settings. I think the work has potential to appeal to a broad audience. That said, I think there is much room for improvement in terms of the theoretical contribution. I also found some of the assumptions contained in the computational models to be potentially invalid in the context of social media behavior. Finally, the paper is missing several key pieces of information about the data that made it difficult to fully evaluate. Below, more details of these concerns.

The main question running through my mind as I read this paper was, why haven’t the authors considered and tested meaningful alternative hypotheses, rather than simply confirming the most obvious one? Are there any reasons to suspect that reinforcement learning on social media might be different than reinforcement learning in other settings? If so, how might it be different? Given that

reinforcement learning is one of the most robust findings in the behavioral science literature – it has been demonstrated across many species and contexts – why is it interesting to test in the context of social media? The main model comparison offered in the paper – a learning model, versus no learning – seems trivial. What aspects of social media might make reinforcement learning in this context unique? Can they test different models that are theoretically informed, in terms of how social media might change the way people learn from rewards? And can the authors find any positive evidence for these differences in the model comparisons?

The authors do consider and attempt to test one interesting possibility: that people may compare the rewards they receive to those of others. However, I found the way that social comparison was operationalized to be somewhat strange. Although it was somewhat difficult to evaluate based on lack of detailed information about the data, it seems like the social comparison models added a term to the prediction error that incorporates the average number of likes per post on the forum in the preceding week. So, the “likes” for a given post get scaled by the average number of likes for all posts on the forum in the past week. In concrete terms, if I post on the forum and get 10 likes, this carries less impact if in the past week, posts on the forum on average got 100 likes, compared with a situation where posts in the past week on average got 5 likes. I question whether this operationalization can accurately capture social comparison effects at the individual user level. This is because, it seems like it would be impossible to tell from the available data, how aware an individual user is of the average number of likes per post on the forum. Let’s say I post on October 1st, I get 5 likes, and then I post a second time on October 10th, and get 10 likes. From October 3rd – 10th, on average posts in the forum are getting 50 likes. Lindstrom’s social comparison model would code my experience as a negative prediction error, since the number of likes I got was lower than the forum average over the past week. But, I haven’t logged in since the 1st. So how would I know that the forum average is 50 likes between the 3rd and the 10th? I am not sure how the authors would be able to determine individual users’ knowledge of whether they are getting more or fewer likes than others in their network. But it seems like it would be necessary to verify this knowledge somehow, in order to be able to make claims about social comparison. To be sure, I do not doubt that social comparison on social media is a real phenomenon. I am just not sure the authors are capturing it in a meaningful way in their data. They also do not provide evidence the social comparison model explains the group level data better than the basic RL model. This may be because they are not measuring social comparison appropriately.

A second concern arises with a key assumption of the authors’ favored model. This model includes an effort cost C that penalizes posting in quick succession. This apparently is imported from animal models of reinforcement learning, where obtaining rewards typically requires physical effort. While the assumption that C is positive makes sense in the context of animal models where effortful physical actions are involved, I am not sure it makes sense in the context of social media. In the supplemental materials, the authors write “posting three times in one day is more costly than posting three times in three days”. This seems like another area where RL in the social media context could be quite different from traditional RL contexts. Logging into social media and “catching up” on the current conversation – particularly in a discussion forum – has a fixed cost (or maybe even a cost that scales positively with time, if it takes longer to catch up the longer you have been offline). Thus, posting three times in quick

succession within the same login session would be expected to actually be less costly than posting three times in three separate login sessions. There may be some nonlinearities in the effort cost function, whereby posting several times in quick succession within the same login is the least costly, and then following that the effort cost is flat. From what I can tell, the authors never tested the possibility that effort costs are negative at very short timescales and/or increasing positively, rather than negatively, with time. More generally, the entire social media industry is designed to make posting as effortless as possible, so this seems like a real opportunity to test precisely how effort costs might differ when RL is put in the social media context.

A third question stems from the authors' decision to limit their analysis of the forum data to posts containing images only. What percentage of all posts do these comprise, i.e., how much of the data was excluded on the basis of this decision? Do the results replicate if the authors consider all posts, not just posts containing images?

Two further questions about timescales that might be theoretically interesting. First, have the authors considered the possibility that "likes" might become less subjectively valuable the less frequently users are posting? Much has been written (albeit not in an academic context) about the false appeal of social media rewards, and how "digital detox" experiences make users realize that social feedback on social media is more hollow than offline experiences. This could be another way that RL differs online vs offline. Relatedly, the authors mention they are examining RL at "timescales rarely or never investigated in the laboratory". It would be great to see a more detailed analysis of what timescales RL is occurring at on social media.

The computational phenotyping was interesting, but difficult to interpret. Do the authors have any theory-driven hypotheses that could guide their analysis of individual differences?

Finally, I found it difficult to evaluate some of the methods and results due to missing information. The authors should report descriptive statistics for all key variables, e.g. to get a sense of what "high" and "low" reward rates mean, in concrete terms. How many likes on average do posts get? How frequently does the average user post? How does that frequency change with every 10% increase in reward rates? It would also be helpful to see the key models written out in the main text rather than the supplement.

Other comments:

-Fig 2C y-axis label is missing

-I didn't have a good sense of how good the model fits were in general. How do they compare, for example, to model fits of RL behavior in lab settings?

-Did models control for whether a given user started a thread?

-SOM p 6, please provide more information about the “linear model” against which the QLE model was compared.

-SOM p 8, reports that outliers falling outside 20th-80th percentiles were removed, but main text reports 10th-90th. Which is correct?

-Fig S2, please include shading of model predictions as in Fig 2C

Reviewer #1 (Remarks to the Author):

Lindstrom and colleagues test whether reinforcement learning mechanisms contribute to behavior on social media. Across a large sample of individuals they apply computational models of reinforcement learning and suggest that such behaviour on social media conforms to principles of reward learning. They also find individual differences in learning are related to a users tendency for social comparison. This is a well written and well-conducted study that aims to answer an important question. I enjoyed reading it and I applaud the authors attempt to explain social media behavior computationally. However, I have some major concerns as to whether the study shows specificity to reinforcement learning mechanisms and the lack of any experimental manipulation.

Response: We sincerely thank the reviewer for the positive comments on our study. As outlined in detail below, we took these comments very seriously and addressed them by (a) conducting comprehensive additional modeling, including models based on foraging theory, and (b) designing and conducting a new behavioral experiment to demonstrate that the manipulation of social reward rate affects response latencies as predicted by our RL account.

Specificity to RL mechanisms – the authors focus their model comparison on two models, an RL model and a model with no learning. However, they do not falsify whether they have simply assessed a limited model space (e.g. Palminteri et al., 2017). It could just be they haven't compared relevant models. Of course this can become a bit intractable as there are potentially unlimited models of behaviour, but there are other plausible models that come to mind.

Response: The reviewer makes the important points that a degree of specificity in models is critical, even if in practice it is impossible to fully ascertain specificity. We completely agree, and thus we took further steps to test the specificity of our model in this revision.

However, before describing these revisions, we would like to note that a key contribution of our study is the proof of principle demonstration that reward learning occurs on social media, regardless of its underlying algorithms—an idea often assumed by never tested. Nonetheless, we have conducted extensive additional model comparisons in order to be as specific as possible. Furthermore, our novel inclusion of an experiment, in which we manipulate average reward rates, provides independent evidence for the proposed reward learning framework.

We summarize the additional model comparison in the main text (page 15):

“Alternative models. To test the specificity of the $\bar{R}L$ model, we compared it with a set of plausible alternative models in the SM. Specifically, we examined models (i) without effort cost (C) or net reward rate (\bar{R}) parameters, (ii) where the effort cost was fixed, or increased (rather than decreased) with post latencies, (iii) without an instrumental response policy, and (iv) based on foraging theory. Each of these provided worse accounts of the data than the $\bar{R}L$ model (see SM, section “*Model comparison with alternative models*”).”

Each of these alternative models were addressed in detail in the Supplementary Material (SM, section “*Model comparison with alternative models*”):

“To assess the specificity of $\bar{R}L$ model, we conducted additional model comparisons that varied key features of the $\bar{R}L$ model. Finally, we compared the $\bar{R}L$ model to a model inspired by foraging theory.

Effect of time dependent terms. The $\bar{R}L$ model explicitly incorporates time-dependent effort and opportunity cost terms (see “*Description of $\bar{R}L$ model*” in main text) that scales with τ_{Post} . To ascertain that these terms, which were based on established theory for free-operant tasks (Niv, Daw, Joel, & Dayan, 2007), contributed to the explanatory power of the $\bar{R}L$ model, we compared it to two alternative models that did not include τ_{Post} - dependent terms. Both alternative models utilized the same policy gradient and likelihood function as the original model, but differed in how prediction errors (equation S3, below) and \bar{R} (equation S5) were computed. Here, we present model RL2:

$$\tau_{POST}^t \sim \exp(Policy^t) \quad [S1]$$

$$\Delta\tau_{Post}^t = \tau_{Post}^t - \tau_{Post}^{t-1} \quad [S2]$$

$$\delta^t = R^t - C - \bar{R}^t \quad [S3]$$

$$Policy^{t+1} = Policy^t + \alpha * \Delta\tau_{Post}^t * \delta^t \quad [S4]$$

$$\bar{R}^{t+1} = \bar{R}^t + \alpha * \delta^t \quad [S5]$$

In model RL3, the effort cost parameter C was removed, which simplifies equation S3 to:

$$\delta^t = R^t - \bar{R}^t \quad [S6]$$

We compared the $\bar{R}L$ model to RL2 and RL3 using AIC_w . The $\bar{R}L$ model provided the best explanation of the data in all four datasets (all exceedance probabilities = 1, see table S3).

Table S3

Dataset	$\bar{\text{RL}}$ model	RL2	RL3
Study 1: Instagram	0.69	0.09	0.22
Study 2: Men’s Fashion	0.68	0.10	0.21
Study 2: Women’s Fashion	0.71	0.15	0.14
Study 2: Gardening	0.76	0.08	0.16

Table S3. Comparison of the $\bar{\text{RL}}$ model to alternative learning models without time-dependent effort- and opportunity cost terms. The table shows the mean AIC_W for each model. The exceedance probability (xp) for the $\bar{\text{RL}}$ model was 1 in all datasets.

Alternative effort cost formulations. The effort cost term of the $\bar{\text{RL}}$ model (main text, eq. 2) is an exponentially decreasing function of τ_{Post} . This formulation is based on established RL theory for free operant tasks (Niv et al., 2007). However, it is possible that the effort cost on social media takes different forms. We evaluated two alternative effort cost formulations: (i) exponentially increasing with time, and (ii) fixed. We did not find any evidence that either a fixed effort cost, or a cost that increased with post latency improved model fit (Table S4).

For the Instagram dataset, the fit of the original $\bar{\text{RL}}$ model and the model with increasing effort cost was similar (direct comparison: $\text{AIC}_W = 0.502$ vs 0.497 , $\text{xp} = .71$). This might be because most users in the Instagram dataset posted with on average short latencies, which would render the influence of a time dependent (either positive or negative) effort cost term less pronounced. In line with this reasoning, we find that the original $\bar{\text{RL}}$ model, with negative effort cost, fits relatively better for users with relatively longer average posting latencies, for whom the difference between the two effort cost variants would be most impactful on model fit (Spearman $\rho = 0.12$, $p < .0001$). In the three other datasets, where average posting latencies were longer (see Table S1), we find that the original $\bar{\text{RL}}$ model, where effort cost decreases with τ_{Post} , best explained the data (see Table S4 below). We also find that the original decreasing effort cost formulation provides the overall best fit when the four datasets are pooled (combined $\text{AIC}_W = 0.46$, t-test against equal weights: $t(4165) = 20.57$, $p < .0001$, $\text{xp} = 1$). Together, these results support the theory-based effort cost formulation of the $\bar{\text{RL}}$ model, but suggest that the effort cost term is not critical for model fit, especially if the average τ_{Post} is short.

Table S4

Dataset	$\bar{\text{RL}}$ model	Fixed cost	Increasing cost
Study 1: Instagram	0.40/0.68	0.20/0	0.40/0.32
Study 2: Men’s Fashion	0.41/1	0.32/0	0.27/0
Study 2: Women’s Fashion	0.61/1	0.22/0	0.17/0
Study 2: Gardening	0.49/1	0.36/0	0.16/0

Table S4. Comparison of the $\bar{R}L$ model to alternative effort cost formulations. See text for details. The table shows the mean AIC_w /exceedance probability for each model and dataset.

Effect of instrumental policy. A key component of the $\bar{R}L$ model is that it allows instrumental learning of the response policy (eq. 2-4, main text). In other words, the model can learn that slower/faster post latencies result in more reward, reflecting the hypothesis that social media users strategically adjust their rate of engagement to maximize social rewards. Alternatively, one can envision a “Pavlovian” policy, where the responses are faster following positive prediction errors (“approach”) and slower following negative prediction errors (“avoidance”). We implemented this “Pavlovian Policy” model by changing the policy update equation (eq. 4, main text) to:

$$Policy^{t+1} = Policy^t + \alpha * -\sqrt[3]{\delta^t} \quad [S7]$$

In this model, the policy is directly updated with the (cube root of) the value prediction error. We take the cube root to reduce the influence of especially large prediction errors on the policy, which preliminary analysis showed was detrimental for model fit. Model comparison showed that the $\bar{R}L$ model explained the data best in all four datasets (t-tests against equal weights, largest p-value = .009, see Table S5). This indicates that people instrumentally learn to maximize rewards by adjusting posting response latencies.

Table S5

Dataset	$\bar{R}L$ model	Pavlovian Policy
Study 1: Instagram	0.62	0.38
Study 2: Men’s Fashion	0.54	0.46
Study 2: Women’s Fashion	0.61	0.39
Study 2: Gardening	0.56	0.44

Table S5. Comparison of the $\bar{R}L$ model to alternative effort cost formulations. See text for details. The table shows the mean AIC_w for each model. The exceedance probability (xp) for the $\bar{R}L$ model was 1 in all datasets.”

In addition, we have developed and tested a model based on foraging theory, as outlined in our next response to the reviewer.

For example, theories of optimal foraging (Charnov, 1976) describe how make foraging decisions are based on the reward rate, the richness of the environment (the average reward rate) and the dispersion of patches. Can the authors show that foraging models, that do not contain a learning component, account for the behavior? I appreciate they ran simulations to show parameter recovery but since the main aim of the paper is to show that social media behavior conforms to RL principles it is very important to show specificity.

Response: We agree with the reviewer that foraging theory is an important theoretical framework for reward guided behavior. However, given the structure of social media (of the type we consider at least), which is so different from typical foraging situations, the application of foraging theory is not completely straight-forward. To our knowledge, no existing foraging model directly corresponds to the structure of social media, such as Instagram, where there arguably are no clear equivalent of patches, no travel time between patches, continuous responding extended over time (instead, posting is binary), et cetera. Furthermore, most of the key variables needed for direct applications of foraging theory (Gabay & Apps, 2020) are unobservable on social media. In contrast, the structure of social media appears more similar to “Skinner boxes”, which originally led to our hypothesis that RL would be a natural framework for understanding social media engagement.

Despite these structural differences, we have put considerable thought and effort into our response to the reviewer’s suggestion. In attempting to address this suggestion, we have therefore developed a new model based on principles of foraging theory.

Briefly, our foraging model (F-model) incorporates the key principle of the marginal value theorem (*to maximize intake rate, a forager should leave a patch when its instantaneous intake rate falls below the long term expected rate in the environment*) (Charnov, 1976). Intuitively, the F-model predicts that one should post when the expected value of posting is higher than the average in the environment during the same time duration. We note that the F-model, as with other mechanistic models of foraging (Constantino & Daw, 2015; Kolling & Akam, 2017), includes a learning component, as the forager’s estimate of the average reward rate needs to be continuously updated.

We fitted this F-model to all four datasets and found that it indeed fits the data better than the baseline model with a constant response policy (“No Learning”), showing that it can account acceptably for social media behavior. However, the F-model fitted the data clearly worse than the RL model. We therefore conclude that RL appears to give a better account of social media behavior than foraging theory, as instantiated in our foraging model. We report the foraging model in the SM (Section “*Model comparison with alternative models*”):

“Comparison with a model based on foraging theory. Foraging theory provides a general framework for how organisms should maximize reward in decision making situations that extend over time (G. H. Pyke, Pulliam, & Charnov, 1977). Although foraging theory is more concerned with deriving optimal decision-making rules than with the precise mechanisms that produce behavior, some studies have successfully compared computational models based on foraging theory and RL (Constantino & Daw, 2015). Following this approach, we developed a stylized model inspired by the principles of foraging theory, in order to assess the specificity of the $\bar{R}L$ model as an explanation of reward maximization on social media. A core principle of foraging theory is that organisms should maximize their net rate of intake by foraging in a given patch until the current reward rate falls below the average in the environment (marginal value theorem (Charnov, 1976)). Because the social media environment involves many unobservables that would be required for a direct application of foraging theory and the marginal value theorem (e.g., travel time, different distinct patches, extended foraging bout (Gabay & Apps, n.d.)), our *F-model* is by necessity relatively abstract. We first describe the model, and then outline the relationship to foraging theory.

The core component of the F-model is the decision to forage (i.e., post) when the expected reward meets or exceeds a threshold T (i.e., $E(R)^t \geq T$). τ_{post} follows an exponential distribution, given by:

$$\tau_{POST}^t \sim \exp(E(R)^t - T = 0) \quad [S8]$$

where T is a free parameter ($0 \leq T \leq \infty$) that determines the threshold. $E(R)^t$ is a linear function of the time since the last post (t_{Last}) and the average reward rate, weighted by a free parameter P ($0 \leq P \leq \infty$). Practically, we solve for t by numerically searching for the root (i.e., 0) of the function $E(R)^t - T = 0$. Intuitively, $E(R)^t$ increases linearly with the time since the last post, with a slope given by the average reward rate:

$$E(R)^t = \bar{R}^t P t_{Last} \quad [S9]$$

The average by unit time reward rate was calculated as a recency-weighted mean, with updating parameter α ($0 \leq \alpha \leq 1$) similar to the $\bar{R}L$ model:

$$\bar{R}^{t+1} = \bar{R}^t + \alpha * \delta^t \quad [S10]$$

$$\delta^t = R^t / \tau_{POST}^t - \bar{R}^t \quad [S11]$$

The F-model is built on the assumption that the expect value of foraging (i.e., posting) goes to 0 directly after a post, and increases with time since the last post. In other

words, likes are assumed to have a refractory time, and this refractory time is dependent on the average reward rate. The assumption that foraging reduces available reward is standard in foraging theory (Graham H Pyke, 1984). The F-model predicts, as the $\bar{R}L$ model, that posting should be more frequent when the reward rate is high, as this maximizes the per unit time incurred reward.

The threshold parameter T can be interpreted in two ways that follow from foraging theory. First, it can be seen as an estimate of the overall average reward in the environment (which we cannot directly observe). Under this interpretation, the F-model decision rule (eq. S8) is equivalent to the marginal value theorem: the forager should “leave” the “patch” (a given social media platform, e.g., Instagram) if the expected value of foraging is lower than the average environmental reward, or conversely, forage in the patch if the expected value is higher than the average environmental value. A second interpretation is based on foraging theory for “sit and wait predators” that forage in one patch (rather than select between patches), and whose prey disperse after a foraging attempt (e.g., a school of small fish) (Hugie, 2003; Katz et al., 2014; Wacht Katz, Abramsky, Kotler, Altstein, & Rosenzweig, 2010). For such predators, the optimal response time is equal to the refractory, or return, time of the prey (Hugie, 2003; Katz et al., 2014; Wacht Katz et al., 2010). Under this interpretation, T reflects the foragers estimated return time, or the time at which available reward (i.e., likes) returns to baseline.

We estimated the F-model, and found that while it provided a better explanation of the data than No Learning, the $\bar{R}L$ model had a better fit in all four datasets (see Table S6, all exceedance probabilities = 1). These results suggest that RL mechanisms provide a preferable account of the temporal dynamics of social media behavior.

Table S6

Dataset	$\bar{R}L$ model	F-model	No Learning
Study 1: Instagram	0.54	0.27	0.18
Study 2: Men’s Fashion	0.47	0.34	0.19
Study 2: Women’s Fashion	0.66	0.18	0.16
Study 2: Gardening	0.6	0.31	0.09

Table S6. Comparison of the $\bar{R}L$ model to the F-model. See text for details. The table shows the mean AIC_w for each model. The exceedance probability (xp) for the $\bar{R}L$ model was 1 in all datasets.”

Machine learning approaches to understand behaviour on the internet are not necessarily new and are already used by many companies to evaluate how to get people to buy products online and personal ad recommendations (e.g. Theocharous et al., 2015, Personalized Ad Recommendation Systems for Life-Time Value Optimization with Guarantees; Kalyanam J, Quezada M, Poblete B, Lanckriet G (2016) Prediction and Characterization of High-Activity Events in Social Media Triggered by Real-World News. PLOS ONE 11(12): e0166694. <https://doi.org/10.1371/journal.pone.0166694>). These papers compare several different models with different parameters. In general, there seems to be a relatively large field of machine learning approaches to social media behaviour that is omitted in the manuscript where lots of different types of sophisticated models beyond simple RL that are used to predict internet behaviour.

Response: We agree with the reviewer that there are existing machine learning papers using RL methods on social media. However, these papers (e.g., the ones cited by the reviewer) do not attempt to explain human behavior and psychology (as we attempt to do), but rather use RL to train systems (e.g., recommender systems) that *interact* with online users/consumers. As such, this literature appears to address research questions very distinct from ours. We have clarified this in the revised introduction, and added one of the references cited by the reviewer (page 4):

“Furthermore, studies that do apply RL approaches to social media data have typically not sought to delineate psychological mechanisms underlying social media use, but instead to optimize systems that interact with users (e.g., by training recommender systems (Theocharous, Research, Thomas, & Ghavamzadeh, 2015)).

We also reference other, more directly relevant papers from the machine learning literature (page 4):

“In addition, results from the few studies that have taken a quantitative approach to human behavior are mixed. In one study, negative evaluation of a post—a type of social punishment—led to deterioration in the quality of future posts, rather than the improvement predicted by learning theory (Cheng, Danescu-Niculescu-Mizil, & Leskovec, 2014). By contrast, in another study, receiving more replies for a post on a specific social media discussion forum predicted a subsequent increase in the time spent on that forum relative to others, consistent with learning theory (Das & Lavoie, 2014).”

Theoretically, I am not sure RL principles to understand behavior on Instagram make much sense. Simple RL models describe how associations between stimuli/actions and outcomes are paired over

time, but in the authors' model they instead use the average reward rate? Or I may have misunderstood something here.

Response: Our application of RL indeed goes beyond the standard in psychology/cognitive neuroscience of modeling discrete choices in the lab. However, our model builds directly on the RL theory developed by Yael Niv and colleagues (Niv, 2007; Niv et al., 2007) to explain rodent behavior in “free operant tasks,” i.e., when the animal not only decides what to do (e.g., lever press), but also decides how often to do it. The same theory has been used to explain how humans adjust the vigor of their actions depending on the reward rate, and how this depends on the level of tonic dopamine (Guitart-Masip, Beierholm, Dolan, Duzel, & Dayan, 2011). We have clarified that our model was based on well-established principles of RL in the revised manuscript by moving the model description to the methods section of the main text (pages 26-30):

“Description of the $\bar{R}L$ model. The $\bar{R}L$ model is a policy gradient version of R-learning (Constantino & Daw, 2015). Rather than storing action values for options and using these as a basis for decision-making, the $\bar{R}L$ model directly updates a parametrized response policy (the mean parameter of an exponential distribution). This is beneficial for learning problems with continuous action spaces (e.g., the latency between responses; (Sutton, Mcallester, Singh, & Mansour, 2000)). In close similarity to standard RL models for discrete action spaces, the $\bar{R}L$ model incrementally learns to adjust its actions (i.e., τ_{Post}) from prediction errors. In contrast to standard RL approaches in psychology, the prediction errors are used to directly adjust the response policy to maximize the undiscounted net rate of reward rather than to update action values.

For each post, the $\bar{R}L$ model selects τ_{Post} as a draw from an exponential distribution, where the mean (i.e., the response policy or threshold) is dynamic:

$$\tau_{POST}^t \sim \exp(Policy^t - \alpha * \bar{R}^t) \quad [1]$$

The initial response policy (i.e. for $t = 1$) was estimated as a free parameter ($0 \leq P \leq \infty$). The subtraction term in eq. 1 implements the momentous effect of the average reward rate (e.g., changes in motivational state), \bar{R}^t , on the response rate, which is independent of learning (Niv et al., 2007). This term, which can be thought of as “Pavlovian” (since it is independent of instrumental behavior), slightly amplifies the effect of the average reward rate on τ_{Post} . Model comparison showed that this term has a small but significant contribution to model fit (results not shown).

The response policy, which determines τ_{Post} , is adjusted based on the prediction error, δ , the difference between the experienced reward (R^t) and the reference level:

$$\delta^t = R^t - \frac{C}{\tau_{Post}^t} - \bar{R}^t * \tau_{Post}^t \quad [2]$$

The reference level explicitly takes into account both the *effort cost* of fast responding and the *opportunity cost* of slow responding (Figure 1 B-C), which is determined by the subjective estimate of the average net reward \bar{R} (Niv et al., 2007; Niv, Joel, & Dayan, 2006). In our application of the model, C ($0 \leq C \leq \infty$) is a user-specific parameter that determines the subjective effort cost of posting (e.g., taking pictures, uploading). This parameter penalizes posting in quick succession (e.g., posting three times in one day is more costly than posting three times in three days; we evaluate different effort cost formulations in section “*Model comparison with alternative learning models*” in the SM). If actions have no intrinsic cost, the optimal policy would be to respond as quickly as possible (Niv, 2007; Niv et al., 2007). The last term in eq. 2 characterizes the opportunity cost of slow responding, which increases with \bar{R} (Figure 1C).

The $\bar{R}L$ model seeks to maximize reward by dynamically updating the response policy by the “gradient”, or slope, of reward. In short, the gradient indicates the direction and distance to the hypothetical maximum net reward that could have been accrued at time t . To compute the gradient, the model tracks the sequential difference between responses (i.e., draws from the policy, which can be thought of as exploration), and combines this quantity with the net reward prediction error (equation 2).

$$\Delta\tau_{Post^t} = \tau_{Post^t} - \tau_{Post^{t-1}} \quad [3]$$

$$Policy^{t+1} = Policy^t + \alpha * \Delta\tau_{Post^t} * \delta^t \quad [4]$$

Thereby, the model can learn the instrumental value of slower vs. faster responses. For example, if $\Delta\tau_{Post^t}$ is positive, which would be the case if the response latency at t was longer than at $t-1$, but δ^t negative, which indicates that the net reward was lower than on average, the gradient (eq. S4) is negative. This results in a reduced response policy (the degree of adjustment depends on the magnitude of both $\Delta\tau_{Post^t}$ and δ^t), and in turn shorter response latencies. In contrast, if both $\Delta\tau_{Post^t}$ and δ^t are either positive or negative, the response policy will increase, which leads to longer response latencies. The average reward rate is updated using the same reward prediction error as the policy, as it directly reflects net reward value (Constantino & Daw, 2015):

$$\bar{R}^{t+1} = \bar{R}^t + \alpha * \delta^t \quad [5]$$

However, regardless of whether slower or faster responses are rewarded, an increase in the average reward rate (eq. 5) results in a higher opportunity cost of time (eq. 2), which in turn results in shorter response latencies (eq. 2, and Figure 1D).

If either $\Delta\tau_{Post^t}$ or $\delta^t = 0$, the model can theoretically be trapped in local reward minima. However, the stochastic policy (eq. 1) ensures that $\Delta\tau_{Post^t}$ is different from 0 with exceedingly high probability, which promotes continuous search by sacrificing convergence (as the model policy will continuously change). In other words, eq. 4 implements an

“optimistic” adjustment of the policy, because it always assumes that the optimal net reward is in the direction of the gradient, and never reached at the current value of the policy. For simplicity, the same step size parameter a ($0 \leq a \leq 1$) was used for all update terms (eq. 1 & 4-5). Inclusion of separate step size parameters for the different update equations did not reliably improve (parameter number penalized) model fit. Additional model information and estimation methods are detailed in the SM (section “*Computational modeling*”).”

I feel that without an experimental manipulation of the participants in the study the conclusions drawn about how social media behavior conforms to RL principles is too much of a leap. The authors could conduct an online study of the participants whose posts they collected and show, for example, that those more sensitive to reward in an independent RL task are those who show more reward sensitive behavior on social media. Or they could set up some kind of simulated Instagram environment where they can control the reward rate and measure the influence on people’s behaviour. I appreciate this is difficult to do but without it the conclusions of the paper are a bit strong I feel.

Response: Following the reviewer’s suggestion, we designed a new experimental paradigm to simulate an Instagram – like environment in order to directly test our model. In similarity to real social media, participants were free to “post” memes (amusing images to share with other users) whenever and as often as they wanted. In accordance with the reviewer's suggestion, we manipulated the reward rate (the number of “likes” participants received per post), and indeed found, as predicted by our reward learning account, that posting response latencies were lower when the reward rate was higher. We hope the reviewer agrees that the addition of this experimental data provides independent and converging evidence for our reward learning account of social media engagement.

We report this experiment in the main text (pages 21-23):

“Experimentally manipulating social reward rates affects posting latencies

Finally, to provide direct evidence that the social reward rate affects posting latencies, we conducted an online experiment in which we experimentally manipulated social rewards and observed posting response latencies. The experiment was designed to capture key aspects of social media, such as Instagram. Participants ($n = 176$) could post “memes”—typically, an amusing image paired with a phrase that is popular on real social media—as often and whenever they wanted during a 25 minute online session (total number of posts = 2206, see SM, section “*Experimental manipulation of social reward rates*” for details). Participants

received feedback on their posts in the form of likes (0-19) from other ostensible online participants (“users”, see Figure S2 for an overview of the experiment). Participants themselves could also indicate “likes” for memes posted by other users. To test whether a higher social reward rate causes shorter response latencies in posting, we increased or decreased the average number of likes participants received between the first and second halves of the session (low reward: 0-9 likes/post, high reward: 10-19 likes/post, drawn from a uniform distribution, with direction of change counter-balanced across subjects). As expected, the average post latency was longer when the social reward rate was lower (0-9 likes/post) relative to higher (10-19 likes/post): $\beta = 0.109$, $SE = 0.044$, $z = 2.47$, $p = .013$ (see Figure 6), corresponding to a 10.9% difference. Notably, participants who reported more followers on Instagram exhibited weaker effects of likes on their behavior (see “*Additional analysis of experiment*” in the SM). This finding parallels how individuals with more Instagram followers in Study 1 exhibited more diminished marginal utility of likes (see “*Study 1*” above and “*Having more followers on Instagram is associated with diminishing marginal utility of likes*” in the SM). These results further support the validity of our experiment in assessing the psychology of real-world social media use. We report additional analyses and robustness checks in the SM (section “*Additional analyses of experiment*”).

To directly relate the experimental results to our model-based analyses of online behavior in Studies 1 and 2, we used the $\bar{R}L$ model to generate subjective \bar{R} time series for the subset of participants with a sufficient number of responses (see SM, section “*Additional analysis of experiment*” for details and additional analyses), and used these (instead of reward condition) to predict response latencies. In accordance with the model fits to the real social media data (Study 1-2), the average response latency was longer when the subjective reward rate was low, relative to high ($n = 156$: $\beta = 0.28$, $SE = 0.045$, $z = 6.24$, $p < .0001$). These experimental results demonstrate that social rewards causally influence response latencies, in support of our conclusion that social reward rates shape real social media behavior.

Figure 6. Experimental manipulation of social reward rates. The estimated effect of social reward rate condition on posting latencies in the online experiment ($n = 176$, see Figure S2 for a design overview). Error-bars denote ± 1 SE of the regression estimates.”

And in the SM (sections “*Experimental manipulation of social reward rates*”):

“**Methods.** We invited participants (with minimum 95% approval rate) on Amazon Mechanical Turk to take part in a study on “humor on social media”. 179 participants completed the study, and were paid 3\$ in compensation. The study was approved by the ethical review board of the University of Amsterdam, The Netherlands. All participants provided informed consent.

Participants were instructed that they would take part in a study of humor on social media, in which they would interact with 19 other online participants (“users”). To resemble the typical structure of social media, the experiment involved a (simulated) “feed”, where the participant could observe images shared by other users, and “post” (share with the other users) their own images (Figure S3). More specifically, the participant could “post” a type of humorous image, known as a “meme” (see Figure S3, A), that is popular on social media (meme images were downloaded from <https://www.reddit.com/r/memes/> and <https://www.reddit.com/r/dankmemes/>, and screened for objectionable content such as profanity, racism, or sexism). As feedback on the posted images, the participant received social feedback (“likes”) from the other users. The participant could also “like” memes posted by other users (Figure S3, B), but to preclude social comparison, they could not observe the likes other users received. Unbeknownst to the participant, the responses of the other were

computer-controlled in order to manipulate the likes provided to the participant (participants were fully debriefed upon the publication of this study).

Importantly, to resemble real social media interaction, the experiment had no trial structure. Instead, the participants could post their own memes and “like” memes posted by the other users whenever and as often they as wished during the experiment (25 minutes). To post a meme image, the participant selected the image they wanted to share with the other users from a set of six randomly selected images (Figure S3, A). The purpose of this was to create a sense of self-expression, while also preventing participants from generating unethical or nonsensical content. Next, the participant was asked (but not required) to provide three “informative and descriptive” nouns or adjectives (termed “tags” to correspond to social media terminology) to the image they selected (e.g., “funny”). The purpose of the tags was to associate an effort cost with posting, as typical on real social media platforms. Next, the participant received 0-19 likes (after waiting 14 s, which matched the display duration of the images that were displayed in the “feed”) in response to their meme post. Crucially, the average amount of likes was manipulated within participant (low reward: $M = 4.5$ likes per post, uniform distribution 0-9. High reward: $M = 14.5$ likes per post, uniform distribution 10-19; low/high order counterbalanced with random assignment to order condition) in order to test the influence of social reward rate on posting response latencies.

After the experiment, the participants were asked to report how many followers they had on Instagram, Twitter, and Facebook, and how many likes they received on average for a post on the social media platform that they typically used. In addition, we administered the nine-item Social Media Disorder Scale (Van Den Eijnden, Lemmens, & Valkenburg, 2016).”

Figure S3. Overview of the experimental design. Online participants could post “memes” (humorous pictures) (A) or “like” the memes posted by other ostensible users (B) whenever and as often they wished during 25 minutes. To post a meme, participants pressed the spacebar and then selected one from a random selection of 6 meme images. Participants were then asked to provide up to 3 “tags” (short keywords) to describe the selected meme. This was included to provide an analogue of the effort cost of posting on social media. Next, participants received feedback (“likes”, represented as filled hearts) on the posted meme. The average amount of likes (4.5 vs 14.5 per post) was manipulated within participant (in counter balanced order across participants) in order to test the influence of social rewards on posting response latencies. The number of likes received for the preceding post was displayed in the simulated “feed.” Note that stimuli are not drawn to scale.

Statistical analysis and data exclusions. We quantified response latencies with the interval from the first opportunity to post (indicated to the participant by the visual prompt “*Press Spacebar to post a Meme*”) until the participant pressed the spacebar (see Figure S3). To analyze the response latencies, we used multilevel GLMMs specifying a gamma distribution with a log link function (using the *glmmTMB* package), as often recommended for response time data (Ng & Cribbie, 2017).

In the main analysis, data were excluded from three participants (out 179) who spontaneously reported that they did not believe the likes were generated by real participants. Additionally, six individual data points with response latencies below 200 ms were excluded, as these are unlikely to reflect real posting decisions. This resulted in a final sample of 176 participants with 2197 responses. Furthermore, we included the (mean-centered) proportion of completed tags per participant (mean proportion = 0.84, median proportion = 1) as an additive covariate, in order to control for effort variability. However, results remained the same regardless of these exclusions or modeling decisions (see “*Additional analysis of experimental results*” below).”

And in the SM (section “Additional analysis of experiment”):

“Robustness analyses. To assess the robustness of the experimental results, we conducted a number of additional analyses. First, we assessed whether the direction with which the reward rate changed (“low to high” vs “high to low”, counterbalanced across subjects) interacted with the reward rate condition, but did not find this to be reliably the case (Order*Reward rate: $\chi^2(1) = 2.18, p = 0.14$).

Second, we evaluated how our data inclusion and exclusion criteria affected the estimated experimental effect. Specifically, we tested whether removing the per participant percentage of completed “tags” covariate, which provides an index of effort cost, from the model affected the estimated effect. This was not the case; the effect of reward condition was almost identical without this covariate (reward condition: $\beta = 0.107, SE = 0.044, z = 2.43, p =$

0.015). Furthermore, we evaluated how the number of posts per participant influenced the estimated effect of reward condition. In the main analysis, we included all participants with at least one post response. To test the robustness of this approach, we performed the same analyses for participants ($N = 156$) with at least 5 posts (reward condition: $\beta = 0.11$, $SE = 0.045$, $z = 2.41$, $p = 0.016$), or with at least one post in each reward condition ($N = 167$, reward condition: $\beta = 0.105$, $SE = 0.044$, $z = 2.40$, $p = 0.016$). Thus, in both cases, the effect of reward condition was comparable to the main analysis. Finally, we tested whether our decisions to remove (i) three participants who reported not believing that likes were generated by real participants, and (ii) six data points with response times shorter than 200 ms affected our results (see “*Experimental manipulation of social reward rates*” for description). We found that neither exclusion criteria had a substantial impact on the estimated experimental effect of reward condition (*i*: $\beta = 0.103$, $SE = 0.044$, $z = 2.33$, $p = 0.019$. *ii*: $\beta = 0.108$, $SE = 0.045$, $z = 2.43$, $p = 0.015$).

Participant-specific reward condition. Due to the random reward distribution and the relatively few responses per participant, the actual difference between high and low reward could differ markedly between individuals, which might lead to imprecise estimates. To account for this possibility, we repeated the analysis with by-participant median likes per reward condition as predictor. We found that using this semi-continuous predictor gave somewhat more precise results ($\beta = -0.062$, $SE = 0.023$, $z = -2.72$, $p = 0.007$) than the categorical reward condition predictor used in the main analyses, which corroborates the conclusion that changes in social reward rate drives changes in response latency.

Analysis of individual differences. To assess if individual differences in self-reported real-life social media behavior moderated the experimental effect of social reward rate on posting response latencies, we analyzed the data from the subset of participants ($n = 145$) who opted to fill in all post experimental questionnaires. We first constructed a full regression model, and then simplified it using backwards elimination to identify the most important moderators. In the full model, we included; (i) the average number of likes received per social media post, (ii) number of followers on Instagram, (iii) number of followers on Twitter, (iv) number of followers on Facebook, and (v) the 9 item version of the *Social Media Disorder Scale* (Van Den Eijnden et al., 2016), all in interaction with reward condition. All continuous predictors were mean-centered. After deletion, the final model included only reward condition (main effect: $\beta = 0.118$, $SE = 0.049$, $z = 2.39$, $p = 0.017$) in interaction with *ii*, the number of followers on Instagram. Intriguingly, a large number of followers on Instagram was associated with a weaker effect of reward condition (Instagram followers * Reward condition interaction: $\beta = -0.0016$, $SE = 0.0006$, $z = -2.62$, $p = 0.009$). This parallels our finding that having more Instagram followers was associated with more strongly diminishing marginal utility of likes in Study 1 (see “*Having more followers on Instagram is associated with*

diminishing marginal utility of likes”). Repeating the analysis with the semi-continuous participant specific median likes predictor (see “*Additional analysis of experiment: Participant-specific reward condition*” above) gave comparable results (Instagram followers * median reward interaction $\beta = 0.0011$, $SE = 0.0003$, $z = 3.54$, $p = 0.0004$).

$\bar{R}L$ model-based analysis. For consistency with our main results, we estimated the $\bar{R}L$ model for participants with at least 5/10 ($n = 156/97$) responses to generate \bar{R} time series (which were converted to High vs Low \bar{R} identically to the main analysis). Naturally, estimating a model with a limited number of data points (e.g., 5-33 in the experiment instead of 10-11649 in Study 1) is prone to high variance and overfitting. Nonetheless, by utilizing the model for latent variable inference (Wilson & Collins, 2019) with multi-level regression (which reduces variance by shrinkage) instead of estimation-based inference, this analysis provides converging evidence that the social reward rate drives behavior, consistent with our theoretical account.

In the main text, we report the direct effect of High vs Low \bar{R} on response latencies (i.e., without the categorical reward condition predictor) for participants with at least 5 responses. By adding both the model-derived and the experiment-based predictor to the model, we next evaluated the shared explanatory value. We found that both the model-derived and experiment-based estimates are reduced in magnitude (0.284 to 0.274, and 0.109 to 0.076, respectively), and that only the model-derived regressor remains conventionally significant (model-derived: $z = 6.0$, $p < .0001$. Experimental: $z = 1.69$, $p = .09$). This indicates that the regressors, as expected, partially explained the same variance, but that the individual specific model-derived regressor better predicts response latencies. Finally, we repeated the analysis for participants with at least 10 responses (as in our analysis of Study 1-2), and find results to be comparable ($n = 97$, $\beta = 0.31$, $SE = 0.052$, $z = 6.18$, $p < .0001$).”

We reference the added experiment throughout the abstract and introduction where appropriate:

“An online experiment ($n = 176$) verified that social rewards causally influence behavior as posited by our computational account.” (abstract)

“We also confirmed, using an online experiment resembling common social media platforms ($n = 176$), that social rewards causally influence behavior as predicted by our computational account.” (page 5)

“Finally, we confirmed the causal influence of the social reward rate on posting response latencies with an online experiment, designed to mimic key aspects of social media platforms.” (page 6)

Minor

The authors should tone down some of the bolder claims in the manuscript as they did not experimentally manipulate any behavior. For example sentences such as ‘Together, our results expand the explanatory reach of learning theory from the behavior of rodents in lab experiments to the behavior of humans on social media’.

Response: We hope the reviewer agrees that our analysis of real social media, together with our new experiment in which we manipulated social reward rates, suggest that learning theory can help explain human behavior on social media.

And ‘the RL model as a compact but rich description (i.e., a computational phenotype) of the mechanisms underlying behavior³⁸.’ There is not any evidence that peoples genotypes are interacting with the environment in the paper, which would be necessary for a computational phenotype.

Response: We referred to “computational phenotypes” as a type of “behavioral phenotype,” and not literally the expression of genes. The concept of a computational phenotype is well established across fields. In our use, we followed the definition of (Patzelt, Hartley, & Gershman, 2018), which does not require interactions of the genotype with the environment: “[...] the concept of a computational phenotype, a collection of parameters derived from computational models fit to behavioral and neural data. This approach represents individuals as points in a continuous parameter space [...] One key advantage of this representation is that it is mechanistic: The parameters have interpretations in terms of cognitive processes, which can be translated into quantitative predictions about future behavior and brain activity”.

Given that the term computational phenotyping is established, we have opted to keep it. We have however toned down the language. We have changed the header of the section (page 19) to “Individual differences in reward learning on social media”. We have also clarified our use of “computational phenotype”, and that we refer to a behavioral phenotype and that we do not measure genotype interactions (page 19):

“[W]e used the parameter estimates of the basic $\bar{R}L$ model as a compact but rich description of the mechanisms underlying behavior—a kind of *computational phenotype* (which is

behavioral in nature and makes no direct reference to the underlying genotype) (Patzelt et al., 2018).”

Reviewer #2 (Remarks to the Author):

In their manuscript, Lindström et al. used computational modeling to examine whether behavior on social media follows classic reward reinforcement learning (RL) patterns. Anonymous data was collected from 4 social media platforms across two studies (Instagram and three blog forums) and was submitted to various learning models. Briefly, the authors found that posting behavior (rate of posting and latency between posts) across the social media platforms was best explained by RL models with parameters that accounted for social rewards ('likes' and comments) on the posts, the effort costs of increased action, the opportunity costs of reduced action, and reward prediction error, consistent with animal models or reward learning. Furthermore, they find that social comparison (e.g., number of 'likes' on one's own post relative to recent posts by others) can explain some of the individual differences in reward learning. This is a timely and well-written manuscript that applies computational affective methodology to ecologically salient behaviors in a uniquely human social setting. The authors convincingly argue their thesis by demonstrating clear and considerate comparisons with other models and they nicely address other possible confounding variables with additional analyses. Overall, the manuscript puts forth novel questions and findings that will be of interest and pertinent across multiple scientific disciplines with important implications that reach beyond academia to a range of industries in society. There are some issues and clarifications that could augment the impact of the paper and are listed below.

Response: We sincerely thank the reviewer for the positive comments about our study and its potential significance.

- The introduction touches on recent assertions that social media engagement follows behavioral patterns that can be compared to addiction. It is important for the authors to qualify in their discussion the ways in which their results not only do not directly provide support for this claim, but also that there are other possible confounding variables at play and other interpretations of the results.

Response: We have taken care to discuss the limitations of our results, as detailed below in response to the reviewer's specific examples, in the revised discussion.

For example, the nature of the feedback in these datasets (other than dichotomous categorization of 'likes' vs comments) is unknown, including the valence (positive/negative) and content (affective/informative, particularly in the forums) of the comments. This is especially critical as negative feedback (a social punishment) may deter social media engagement as referenced in Cheng et al., 2014. As another example, social rewards may or may not motivate other social media behaviors (e.g., viewing others' posts, 'liking' and commenting on others' posts), which are not accounted for in this study.

Response: We agree with the reviewer that there are multiple factors in addition to “likes” that can influence behavior on social media, and that our analysis is limited to one type of behavior (i.e., posting). However, to clarify, the feedback was always in the form of “likes,” and likes always represent positive feedback. We thus agree that extending our modeling approach to negative feedback (e.g., Reddit) would be very informative. In the revised discussion section (page 27), we clarified the limitations of focusing on likes and posting and added discussion on the importance of also considering social punishments in future work:

“Furthermore, while our research focused on the effect of positive social rewards (i.e., likes) on posting behavior, negative feedback, which is rampant on many social media platforms (e.g., down votes), is also likely to drive learning. The RL framework we proposed here may also be extended to include such social punishments. For example, treating social punishments as reinforcement with negative utility would, in principle, allow direct application of the $\bar{R}L$ model in its current form, but it is possible that additional motivational factors, such as negative reciprocity, also play an important role in aversely motivated social media behavior. In addition, as we focused on the timing, rather than the content (e.g., of images or comments), of social media posts, an important future goal will be to characterize how people learn to produce actions (e.g., posting content, comments on others' posts) that maximize reward. Our $\bar{R}L$ model could be modified to include action selection as a part of the response policy (Niv et al., 2007).”

Furthermore, it is unknown based on these results whether RL-based behaviors on social media have adaptive or maladaptive physical and psychological outcomes, which would speak more directly to the external claims that social media behavior follows patterns similar to addiction.

Response: We agree with the reviewer that our results do not speak to the adaptive or maladaptive consequences of social media use, which we have clarified in the discussion (page 25):

“Although our data do not speak directly to whether intense social media use is maladaptive or addictive in nature, it suggests a new approach for asking such critical questions.”

- Another important confounding variable that should be discussed is age of the users. Instagram, for example, is wildly popular among adolescents (ages 13-17) and emerging adults (ages 18-24). Relatedly, these populations have also been shown to be more sensitive to social rewards³. Do the authors have access to this information in any dataset? It would be interesting information to add if available. It would behoove the authors to discuss the possible effects of user age as it relates to their findings and broader implications of this work.

Response: Although a consideration of participant age in reward learning is interesting, it would be a misnomer to call this a confound, as age effects would not provide an alternative to our main conclusions. Nevertheless, we agree with the reviewer that the age of social media users often skews young (as in Instagram) and that this could limit the generalizability of our findings. Although we do not have any information about age in any of the datasets we analyze, unfortunately, the subject matter of the forums examined in Study 2—gardening and fashion (especially expensive adult-focused fashion) suggest an age range beyond adolescence. Nevertheless, we cannot make a quantitative statement about the age distribution. We have added a discussion note about user age and the importance of social rewards in adolescence to the Discussion section of the revised manuscript (page 25):

“Our results raise several new questions regarding the role of reward in social media behavior. First, while our analysis of anonymous, real-world social media data precluded demographic characterization of users, it is possible that certain demographic factors, such as age, may moderate the effect of reward learning in online behavior. For example, adolescents tend to be more sensitive than adults to social rewards and punishments (Crone & Konijn, 2018), and thus our results may be particularly informative to questions of adolescent social media behavior. An examination of the developmental trajectories of the computational phenotypes in social reward learning identified here, and their relations to individual differences in psychological traits, could further illuminate age effects in online behavior.

- It would be beneficial to the reader if the authors were to report descriptive statistics for their subjects and primary variables of interest (e.g., how many followers the users have, how often they post, how much feedback was given across the different platforms).

Response: We agree with the reviewer that this information is beneficial for the reader, and apologize for omitting in in the original submission. We have added these descriptive statistics for all datasets to Table S1 in the supplementary information:

” Table S1

Dataset	Likes				τ_{Post}				Followers	
	M Pop.	Md Pop.	M Ind.	Md Ind.	M Pop.	Md Pop.	M Ind.	Md Ind.	M Pop.	Md Pop.
Study 1: Instagram	377	43	338	74	1.82	0.95	3.12	1.45	324	145
Study 2: Men’s Fashion	17.4	14	16.46	12	14.61	2.99	33.64	6.99	-	-
Study 2: Women’s Fashion	6									
Study 2: Garden	16.4	16	17.22	16.5	11.96	1.75	24.47	4.55	-	-
	5									
	6.1	4	5.43	4	8.03	1.01	22.2	2.98	-	-

Table S1. Descriptive statistics for likes, τ_{Post} , and followers across the four datasets. The table shows both sample averages (Pop.), and individual-level averages (Ind.). The latter was generated by computing the mean (M) and the median (Md) for each individual, and then summarized with mean-of-means and median-of-medians of the individual level averages.

We reference the new table S1 on page 6-7 in the main text.

- The RL model had worse fit for individuals with many followers on Instagram. The authors posit that individuals with larger followings may be motivated by monetary rewards, and these rewards often only come into play when individuals have followings between 5,000 and 100,000+ by some estimates^{4,5} (hence one reason why it’d be helpful to have descriptive statistics on these users for comparison). Another possible interpretation of these findings is that individuals with larger followings, on average, receive consistently higher amounts social rewards (e.g., ‘likes’), and have therefore become habituated to the feedback.

Response: We thank the reviewer for making this important point. To assess whether individuals with more followers might have habituated to social rewards, and thereby required more likes for a given level of subjective value, we augmented our basic $\bar{R}L$ model with different “utility functions.” Notably, and in line with the reviewer’s hypothesis, we found that (i) the data in Study 1 (Instagram),

but not in Study 2, is explained better by including a term for non-linear diminishing marginal utility of likes, and (ii) that individuals with many followers exhibit more strongly diminishing marginal utility of likes. These results are consistent with the reviewer's suggestion, which we now discuss in the manuscript.

We report these new analysis in the main text (page 12):

“Interestingly, we found that individuals with more Instagram followers exhibited non-linearly diminishing subjective value (utility) of likes or, in other words, derived less subjective value from each like (see SM, section “*Having more followers on Instagram is associated with diminishing marginal utility of likes*”). This suggests that individuals with many followers might habituate to likes”.

And the Supplementary Material (section “*Having more followers on Instagram is associated with diminishing marginal utility of likes*”)

“In the basic $\bar{R}L$ – model, we assumed that the utility of likes followed an identity function (i.e., $u(R) = R$). However, this might not be the case on social media such as Instagram, where posts can garner thousands of likes which, in turn, might make users insensitive or habituated to social rewards.

In order to investigate the utility function of likes on social media in more detail, we assessed three different, nested utility functions for R : (i) $u(R) = sR^d$, (ii) $u(R) = R^d$, (iii) $u(R) = sR$, where s ($0 \leq s \leq \infty$) and d ($0 \leq d \leq \infty$) are free parameters added to the $\bar{R}L$ – model. As evident, in formulations *i-ii*, the utility of R followed a nonlinear function that saturates more quickly for lower values of d , while *iii* simply scales R with s . Model comparison showed that model *ii*, which we refer to as the $\bar{R}L^d$ - model, had the best fit (AIC_w : $i = 0.15$, $ii = 0.51$, $iii = 0.34$), and that the difference was unlikely to be due to chance (one sample t-test against $P = 0.34$: $t(2038) = 22.0$, $p < .0001$, $xp = 1$). Notably, this formulation entails strongly diminishing marginal utility of R for values below 1 (1 = no discounting). For example, if an individual with $d = 0.8$ receives 100 likes, the effective utility is ~ 40 , while for 1000 likes, the utility is ~ 250 . The mean d was reliably below 1 ($M = 0.9$, $t(2038) = 39.4$, $p < .0001$), and with a skewed distribution (25th quantile = 0.006, 50th = 0.77, 75th = 1.35, 100th = 6.9), meaning that for most individuals, the marginal utility of likes was strongly diminishing.

Adding the non-linear utility function $u(R) = R^d$ to the basic $\bar{R}L$ – model better accounted for the data (AIC_w : 0.59 vs 0.41, $x_p = 1$).

The relative improvement in model fit was positively associated with Instagram follower number (linear regression of log follower number on the difference in AIC_w relative to No Learning between $\bar{R}L^d$ and $\bar{R}L$ – models: $\beta = 0.013$, $SE = 0.003$, $t = 4.07$, $p < .0001$. Regression of log follower number on AIC_w directly comparing $\bar{R}L^d$ and $\bar{R}L$ – models: $\beta = 0.011$, $SE = 0.004$, $t = 2.94$, $p = .003$), indicating that incorporating diminishing marginal utility of likes improved model fit particularly for individuals with many followers. This suggests that individuals with many followers might have more strongly diminishing marginal utility functions. We tested this notion directly by predicting (log-transformed) follower number from the (standardized) estimated d parameter using linear regression (estimates were very similar when using negative-binomial regression, and when including all model parameters), controlling for differences in the number of posts (which otherwise might confound the relationship). We found that a lower estimated d parameter (i.e., more strongly diminishing marginal utility) was associated with a higher number of followers ($\beta = -0.19$, $SE = 0.044$, $t = -4.16$, $p < .0001$). In other words, the marginal utility of likes, which drives learning, was lower for individuals with more followers. Together, these results show that individuals with many followers might habituate to social rewards, necessitating more likes for an equivalent motivational effect.

In contrast, we found no evidence for diminishing marginal utility in Study 2, as the basic $\bar{R}L$ – model fit best in all three datasets (see Table S2). This is likely due to the, on average, much lower number of likes per post in Study 2 than Study 1 (see Table S1).

Table S2

Dataset	$u(R) = R$	$u(R) = sR^d$	$u(R) = R^d$	$u(R) = sR$
Men's Fashion	0.42	0.15	0.26	0.15
Women's Fashion	0.46	0.08	0.27	0.19
Gardening	0.37	0.12	0.25	0.26

Table S2. Comparison of utility functions in Study 2 showed, in contrast to Study 1, no evidence for diminishing marginal utility of likes in reinforcement learning. The table shows the relative evidence (AIC_w) for different utility functions added to the $\bar{R}L$ – model. $u(R) = R$ denotes the standard $\bar{R}L$ – model. The exceedance probability (x_p) for $u(R) = R$ was 1 in all Study 2 datasets.”

- That the data collected on Instagram originated from individuals who partook in a contest should also be further discussed. It is unclear whether these individuals partook in this and other contests often, and how their participation in these online contests might affect their posting behavior.

Response: The reviewer makes an important point. We therefore analyzed whether contest participation was related to posting behavior and found that it was not. We now report additional information about contest participation and relation to posting behavior in the Supplementary Material (section “*Additional information about Study 1*”):

“Inclusion in the original Instagram dataset (collected by (Ferrara, Interdonato, & Tagarelli, 2014)) was based on participation in at least one of Instagram’s weekly photography contests. Contest participation was denoted by the addition of a hashtag with prefix “#whp-“ to an Instagram post. All media uploaded by a random selection of 2,100 users with at least one “#whp-“ hashtag (including media that were not tagged with #whp-hashtags) were gathered and their information retrieved and stored. Study 1 was based on a subset (with at least 10 posts, $n = 2,039$) of these users.

To quantify contest participation in our dataset, we compared the number of posts with a “#whp-“ hashtag to the total number of posts, and found that “#whp-“ hashtags comprised 2.3% of the total number of posts. The number of “#whp-“ hashtags per user ranged from 1 to 275, with a median of 19 (comprising 0.0005 % to 71 % of posts). We show below that the number of contest participations did not predict social media behavior (“*No evidence for association between Instagram photo contest participation and social media behavior*”).

And section “*No evidence for association between Instagram photo contest participation and social media behavior*”:

“Because the Instagram dataset used in Study 1 was based on participation in a photography contest on Instagram (see “*Additional information about Study 1*” for details), we tested whether the number of such contest participations, as indexed by the number of posts with “#whp-“ hashtags, was associated with social media behavior (as measured by our $\bar{R}L$ model). We predicted (log) “#whp-“ tag number from the estimated parameters of the $\bar{R}L$ model, together with the total number of posts (which naturally is the most important predictor) using linear regression. We found no evidence that the number of “#whp-“ tags was associated with $\bar{R}L$ parameters (neither using the basic $\bar{R}L$ model or the $\bar{R}L$ model augmented with a non-linear utility function): the lowest p -value for any estimated parameter was ~ 0.25 . Together,

these results indicate that contest participation did not have any clear influence on posting behavior, and thus our results are likely to generalize beyond this context (as further suggested by Study 2 results).

We briefly reference these sections from the main text (pages 26):

“Study 1 was based on data from a previously published study (see (Ferrara et al., 2014) for further information), in which data collection was based on a random sample of individuals who partook in a specific photography contest on Instagram in 2014. We find no evidence that contest participation was related to posting behavior (see SM, section “*Additional information about Study 1*”).

- It would be useful for the authors to comment further on the implications of the 4 computational phenotypes. What might attribute to such individual differences? What might this suggest about social media engagement on a broader level?

Response: We agree with the reviewer that the ontogeny, and consequences, of different computational phenotypes is a very interesting question. However, the computational phenotyping analysis was data-driven and exploratory rather than hypothesis testing. Thus, future work is needed for relating the different phenotypes to external measures, both pertaining to social media per se (e.g., risk for social media addiction), and to psychological traits (e.g., personality). Furthermore, our results might, for example, serve as basis for developing questionnaires about social media use (where one might expect 4 factors corresponding to the 4 phenotypes), and for inspiring different types of “interventions” for problematic social media use (e.g., where costs and rewards are differentially manipulated for different phenotypes).

We have clarified the data-driven nature of the computational phenotyping analysis in the revised manuscript (page 20):

“[W]e used the three parameters of the original $\bar{R}L$ model (which is nested in the model variations estimated for Study 1 & 2), estimated for each individual from datasets 1-4 (total $n = 4,168$), as input for k-means clustering, an unsupervised, data-driven method for finding sub-groups in multidimensional data.”

We have also clarified, within the context of adolescent social media use, that additional work is needed for understanding the development of differing computational phenotypes (page 25):

“[A]dolescents tend to be more sensitive than adults to social rewards and punishments (Crone & Konijn, 2018), and thus our results may be particularly informative to questions of adolescent social media behavior. An examination of the developmental trajectories of the computational phenotypes in social reward learning identified here, and their relations to individual differences in psychological traits, could further illuminate age effects in online behavior.

- Regarding the social comparison model, it would be helpful to clarify how the current model differs from models such as the Fehr-Schmidt inequity aversion model. Also, how were disadvantageous and advantageous inequality assigned (based on post content, etc?)

Response: Inequality was defined based on comparison of one’s own outcomes (i.e., likes) and the average amount of likes on the forum (at that time point). We found that a model with only a form of disadvantageous inequality, where the average likes others receive effectively reduce (i.e., subtract from) the value of one’s own outcome, best accounted for the data. For example, if a user received 35 likes for a post, and the average amount of likes other users received was 40, the effective utility (which is used for computing the value prediction error) would be -5 (note that in the actual social comparison model, a free parameter regulated the strength of social comparison).

The Fehr-Schmidt (F-S) model describes decisions made in social interactions, which affects both the focal individual and others. In the F-S model, both advantageous and disadvantageous inequality is negative, and reduces the utility of one’s own outcomes. This makes sense in constrained interactions, such as the ultimatum game, but less so on social media where the outcomes one individual receive will be relatively independent of the outcomes others receive. In other words, social media users are typically not making any decisions that obviously influence the likes both they and other users receive. For this reason, our social comparison models (the main model and the alternatives we describe in the Supplementary Material) instead implement upwards social comparison (i.e., disadvantageous inequality) as reducing the utility of one’s outcomes, and downward social comparison (advantageous inequality) as increasing the utility of ones outcomes. To ascertain that this assumption was not mistaken, for completeness we fitted the F-S model (i.e., where both forms of inequality reduces utility) to the Men’s Fashion dataset (for brevity). As expected, we found that the F-S model provided a much worse fit than our social comparison models (median AIC, lower is better: F-S = 239.8, upward+downward comparison = 223.5, only upwards comparison [our main social comparison model] = 217.8).

Reviewer #3 (Remarks to the Author):

Lindstrom and colleagues present data testing a hypothesis that reinforcement learning mechanisms contribute to the social media behavior. Using data from Instagram and online forums, they show that the frequency of posting is related to the number of “likes” received for posts. They fit a variety of reinforcement learning models to the data and evaluate whether there are distinct “computational phenotypes” occupying different regions of parameter space in the RL models. They further consider how the distribution of computational phenotypes can be explained by individual differences in sensitivity to social comparison. The authors conclude their findings “offer new insights into this emergent mode of human behavior on an unprecedented scale. There is a lot to like about this paper. Fitting computational models to behavior on social media is novel and interesting. I appreciated the authors’ efforts to link their approach with animal models, and the replications across multiple social media settings. I think the work has potential to appeal to a broad audience. That said, I think there is much room for improvement in terms of the theoretical contribution. I also found some of the assumptions contained in the computational models to be potentially invalid in the context of social media behavior. Finally, the paper is missing several key pieces of information about the data that made it difficult to fully evaluate. Below, more details of these concerns.

Response: We sincerely thank the reviewer for the positive comments on our study. As outlined in detail below, we have addressed the reviewers concerns by conducting additional modeling and analysis, and added the information found to be missing in the original submission.

The main question running through my mind as I read this paper was, why haven’t the authors considered and tested meaningful alternative hypotheses, rather than simply confirming the most obvious one? Are there any reasons to suspect that reinforcement learning on social media might be different than reinforcement learning in other settings? If so, how might it be different?

Given that reinforcement learning is one of the most robust findings in the behavioral science literature – it has been demonstrated across many species and contexts – why is it interesting to test in the context of social media? The main model comparison offered in the paper – a learning model, versus no learning – seems trivial.

Response: In this comment, the reviewer is essentially suggesting that because our theoretical account seems obvious, it isn’t worth testing with data. We see it the other way: when an explanation becomes extremely popular despite no evidence, it’s crucial to conduct a serious test. We tested the hypothesis that social media engagement is underpinned by reinforcement learning (RL) because this notion has considerable popularity in both popular press and in popular science, without any direct

empirical backing. For example, the Google search "social media" + "skinner box" gives 25500 results. Despite the apparent popularity of this idea, we could not identify *any single study* that provided direct evidence for the involvement of RL mechanisms in social media engagement. As we all know, popular belief can be wrong and needs to be put to test. Moreover, as social media is such a key area for modern (social) behavior, testing the mechanisms of how it is used is an important goal. We contrast an RL model against No Learning just for this reason – to test whether RL can account for social media engagement. Thereby, we could exclude the, a priori just as likely, alternative that people post for other reasons (e.g., when they have experienced something they want to share). Yet, by explaining also *when* people post, our model-based insights go well beyond showing that RL is at play. In fact, it was not a priori evident that RL could account for post latencies on social media, where the relevant time-scales are days and weeks. More generally, our results show that evolutionary old principles of reward learning, which allows successful foraging in interaction with nature, can help explain human behavior in a novel environment (social media) that our ancestors had no experience with. Again, this was not a priori evident.

Nevertheless, we agree with the reviewer that we could take our work to the next level by further disentangling the underlying computational mechanisms. To this end, we have conducted extensive additional model comparison (we discuss some of these comparisons in detail in other responses to the reviewer, below). These results are summarized on page 15:

“Alternative models. To test the specificity of the $\bar{R}L$ model, we compared it with a set of plausible alternative models in the SM. Specifically, we examined models (i) without effort cost (C) or net reward rate (\bar{R}) parameters, (ii) where the effort cost was fixed, or increased (rather than decreased) with post latencies, (iii) without an instrumental response policy, and (iv) based on foraging theory. Each of these provided worse accounts of the data than the $\bar{R}L$ model (see SM, section “*Model comparison with alternative models*”).”

And in the Supplementary Material (section “*Model comparison with alternative models*”):

“To assess the specificity of the $\bar{R}L$ model, we conducted additional model comparisons that varied key features of the $\bar{R}L$ model. Finally, we compared the $\bar{R}L$ model to a model inspired by foraging theory.

Effect of time dependent terms. The $\bar{R}L$ model explicitly incorporates time-dependent effort and opportunity cost terms (see “*Description of $\bar{R}L$ model*” in main text) that scales with τ_{Post} . To ascertain that these terms, which were based on established theory for free-operant tasks (Niv et al., 2007), contributed to the explanatory power of the $\bar{R}L$ model, we compared it to

two alternative models that did not include τ_{post} - dependent terms. Both alternative models utilized the same policy gradient and likelihood function as the original model, but differed in how prediction errors (equation S3, below) and \bar{R} (equation S5) were computed. Here, we present model RL2:

$$\tau_{POST^t} \sim \exp(Policy^t) \quad [S1]$$

$$\Delta\tau_{Post^t} = \tau_{Post^t} - \tau_{Post^{t-1}} \quad [S2]$$

$$\delta^t = R^t - C - \bar{R}^t \quad [S3]$$

$$Policy^{t+1} = Policy^t + \alpha * \Delta\tau_{Post^t} * \delta^t \quad [S4]$$

$$\bar{R}^{t+1} = \bar{R}^t + \alpha * \delta^t \quad [S5]$$

In model RL3, the effort cost parameter C was removed, which simplifies equation S3 to:

$$\delta^t = R^t - \bar{R}^t \quad [S6]$$

We compared the $\bar{R}L$ model to RL2 and RL3 using AIC_w . The $\bar{R}L$ model provided the best explanation of the data in all four datasets (all exceedance probabilities = 1, see table S3).

Table S3

Dataset	$\bar{R}L$ model	RL2	RL3
Study 1: Instagram	0.69	0.09	0.22
Study 2: Men's Fashion	0.68	0.10	0.21
Study 2: Women's Fashion	0.71	0.15	0.14
Study 2: Gardening	0.76	0.08	0.16

Table S3. Comparison of the $\bar{R}L$ model to alternative learning models without time-dependent effort- and opportunity cost terms. The table shows the mean AIC_w for each model. The exceedance probability (xp) for the $\bar{R}L$ model was 1 in all datasets.

Alternative effort cost formulations. The effort cost term of the $\bar{R}L$ model (main text, eq. 2) is an exponentially decreasing function of τ_{post} . This formulation is based on established RL theory for free operant tasks (Niv et al., 2007). However, it is possible that the effort cost on social media takes different forms. We evaluated two alternative effort cost formulations: (i) exponentially increasing with time, and (ii) fixed. We did not find any evidence that either a fixed effort cost, or a cost that increased with post latency improved model fit (Table S4). For the Instagram dataset, the fit of the original $\bar{R}L$ model and the model with increasing effort cost was similar (direct comparison: $AIC_w = 0.502$ vs 0.497 , $xp = .71$). This might be because most users in the Instagram dataset posted with on average short latencies, which would render the influence of a time dependent (either positive or negative) effort cost term less pronounced. In line with this reasoning, we find that the original $\bar{R}L$ model, with negative

effort cost, fits relatively better for users with relatively longer average posting latencies, for whom the difference between the two effort cost variants would be most impactful on model fit (Spearman $\rho = 0.12$, $p < .0001$). In the three other datasets, where average posting latencies were longer (see Table S1), we find that the original $\bar{R}L$ model, where effort cost decreases with τ_{post} , best explained the data (see Table S4 below). We also find that the original decreasing effort cost formulation provides the overall best fit when the four datasets are pooled (combined $AIC_w = 0.46$, t-test against equal weights: $(t(4165) = 20.57, p < .0001, xp = 1)$). Together, these results support the theory-based effort cost formulation of the $\bar{R}L$ model, but suggest that the effort cost term is not critical for model fit, especially if the average τ_{post} is short.

Table S4

Dataset	$\bar{R}L$ model	Fixed cost	Increasing cost
Study 1: Instagram	0.40/0.68	0.20/0	0.40/0.32
Study 2: Men’s Fashion	0.41/1	0.32/0	0.27/0
Study 2: Women’s Fashion	0.61/1	0.22/0	0.17/0
Study 2: Gardening	0.49/1	0.36/0	0.16/0

Table S4. Comparison of the $\bar{R}L$ model to alternative effort cost formulations. See text for details. The table shows the mean AIC_w /exceedance probability for each model and dataset.

Effect of instrumental policy. A key component of the $\bar{R}L$ model is that it allows instrumental learning of the response policy (eq. 2-4, main text). In other words, the model can learn that slower/faster post latencies result in more reward, reflecting the hypothesis that social media users strategically adjust their rate of engagement to maximize social rewards. Alternatively, one can envision a “Pavlovian” policy, where the responses are faster following positive prediction errors (“approach”) and slower following negative prediction errors (“avoidance”). We implemented this “Pavlovian Policy” model by changing the policy update equation (eq. 4, main text) to:

$$Policy^{t+1} = Policy^t + \alpha * -\sqrt[3]{\delta^t} \quad [S7]$$

In other words, the policy is directly updated with the (cube root of) the value prediction error. We take the cube root to reduce the influence of especially large prediction errors on the policy, which preliminary analysis showed was detrimental for model fit. Model comparison showed that the $\bar{R}L$ model explained the data best in all four datasets (t-tests

against equal weights, largest p-value = .009, see Table S5). This indicates that people learn to maximize rewards by adjusting posting response latencies.

Table S5

Dataset	$\bar{R}L$ model	Pavlovian Policy
Study 1: Instagram	0.62	0.38
Study 2: Men’s Fashion	0.54	0.46
Study 2: Women’s Fashion	0.61	0.39
Study 2: Gardening	0.56	0.44

Table S5. Comparison of the $\bar{R}L$ model to alternative effort cost formulations. See text for details. The table shows the mean AIC_w for each model. The exceedance probability (xp) for the $\bar{R}L$ model was 1 in all datasets.

Comparison with a model based on foraging theory. Foraging theory provides a general framework for how organisms should maximize reward in decision making situations that extend over time (G. H. Pyke et al., 1977). Although foraging theory is more concerned with deriving optimal decision-making rules than with the precise mechanisms that produce behavior, some studies have successfully compared computational models based on foraging theory and RL (Constantino & Daw, 2015). Following this approach, we developed a stylized model inspired by the principles of foraging theory, in order to assess the specificity of the $\bar{R}L$ model as an explanation of reward maximization on social media. A core principle of foraging theory is that organisms should maximize their net rate of intake by foraging in a given patch until the momentous reward rate falls below the average in the environment (marginal value theorem(Charnov, 1976)). Because the social media environment involves many unobservables that would be required for a direct application of foraging theory and the marginal value theorem (e.g., travel time, different distinct patches, extended foraging bout (Gabay & Apps, n.d.)), our *F-model* is by necessity relatively abstract. We first describe the model, and then outline the relationship to foraging theory.

The core component of the F-model is the decision to forage (i.e., post) when the expected reward meets or exceeds a threshold T (i.e., $E(R)^t \geq T$). τ_{post} follows an exponential distribution, given by:

$$\tau_{POST}^t \sim \exp(E(R)^t - T = 0) \quad [S8]$$

where T is a free parameter ($0 \leq T \leq \infty$) that determines the threshold. $E(R)^t$ is a linear function of the time since the last post (t_{Last}) and the average reward rate, weighted by a free parameter P ($0 \leq P \leq \infty$). Practically, we solve for t by numerically searching for the root

(i.e., 0) of the function $E(R)^t - T = 0$. Intuitively, $E(R)^t$ increases linearly with the time since the last post, with a slope given by the average reward rate:

$$E(R)^t = \bar{R}^t P t_{Last} \quad [S9]$$

The average by unit time reward rate was calculated as a recency-weighted mean, with updating parameter α ($0 \leq \alpha \leq 1$) similar to the $\bar{R}L$ model:

$$\bar{R}^{t+1} = \bar{R}^t + \alpha * \delta^t \quad [S10]$$

$$\delta^t = R^t / \tau_{POST} - \bar{R}^t \quad [S11]$$

The F-model is built on the assumption that the expect value of foraging (i.e., posting) goes to 0 directly after a post, and increases with time since the last post. In other words, likes are assumed to have a refractory time, and this refractory time is dependent on the average reward rate. The assumption that foraging reduces available reward is standard in foraging theory (Graham H Pyke, 1984). The F-model predicts, as the $\bar{R}L$ model, that posting should be more frequent when the reward rate is high, as this maximizes the per unit time incurred reward.

The threshold parameter T can be interpreted in two ways that follow from foraging theory. First, it can be seen as an estimate of the overall average reward in the environment (which we cannot directly observe). Under this interpretation, the F-model decision rule (eq. S8) is equivalent to the marginal value theorem: the forager should “leave” the “patch” (a given social media platform, e.g., Instagram) if the expected value of foraging is lower than the average environmental reward, or conversely, forage in the patch if the expected value is higher than the average environmental value. A second interpretation is based on foraging theory for “sit and wait predators” that forage in one patch (rather than select between patches), and whose pray disperse after a foraging attempt (e.g., a school of small fish) (Hugie, 2003; Katz et al., 2014; Wacht Katz et al., 2010). For such predators, the optimal response time is equal to the refractory, or return, time of the prey (Hugie, 2003; Katz et al., 2014; Wacht Katz et al., 2010). Under this interpretation, T reflects the foragers estimated return time, or the time at which available reward (i.e., likes) returns to baseline.

We estimated the F-model, and found that while it provided a better explanation of the data than No Learning, the $\bar{R}L$ model had a better fit in all four datasets (see Table S6, all exceedance probabilities = 1). These results suggest that RL mechanisms provides a fitting account of the temporal dynamics of social media behavior.

Table S6

Dataset	$\bar{R}L$ model	F-model	No Learning
Study 1: Instagram	0.54	0.27	0.18
Study 2: Men’s Fashion	0.47	0.34	0.19
Study 2: Women’s Fashion	0.66	0.18	0.16

Study 2: Gardening	0.6	0.31	0.09
-----	------	------

Table S6. Comparison of the $\bar{R}L$ model to the F-model. See text for details. The table shows the mean AIC_w for each model. The exceedance probability (xp) for the $\bar{R}L$ model was 1 in all datasets.”

What aspects of social media might make reinforcement learning in this context unique? Can they test different models that are theoretically informed, in terms of how social media might change the way people learn from rewards? And can the authors find any positive evidence for these differences in the model comparisons?

Response: Rather than assuming that RL on social media is unique, our hypothesis was that it follows the same principles as RL in other situations. For this reason, we based our model on RL theory developed for rodents lever pressing for primary rewards. The fact that our model provides a good account of social media engagement supports this hypothesis.

However, we have identified two ways in which RL on social media appears to be unique. First, we provide positive evidence for social comparison, as outlined in our next response to the reviewer.

Second, we find that individuals with more followers on Instagram (Study 1) exhibit non-linearly diminishing marginal utility of likes, which is founded in economics. In other words, it seems as if people with many followers habituate to likes, and require more likes to reach the same subjective value. We report this new analysis in the main text (pages 12):

“Interestingly, we found that individuals with more Instagram followers exhibited non-linearly diminishing subjective value (utility) of likes, or in other words, derived less subjective value from each like (see SM, section “*Having more followers on Instagram is associated with diminishing marginal utility of likes*”). This suggests that individuals with many followers might habituate to likes”.

And the Supplementary Material (section “*Having more followers on Instagram is associated with diminishing marginal utility of likes*”)

“In the basic $\bar{R}L$ – model, we assumed that the utility of likes followed an identity function (i.e., $u(R) = R$). However, this might not be the case on social media such as Instagram, where

posts can garner thousands of likes which, in turn, might make users insensitive or habituated to social rewards.

In order to investigate the utility function of likes on social media in more detail, we assessed three different, nested utility functions for R : (i) $u(R) = sR^d$, (ii) $u(R) = R^d$, (iii) $u(R) = sR$, where s ($0 \leq s \leq \infty$) and d ($0 \leq d \leq \infty$) are free parameters added to the $\bar{R}L$ – model. As evident, in formulations *i-ii*, the utility of R followed a nonlinear function, which saturates more quickly lower values of d , while *iii* simply scales R with s . Model comparison showed that model *ii*, which we refer to as the $\bar{R}L^d$ - model, had the best fit (AIC_W : $i = 0.15$, $ii = 0.51$, $iii = 0.34$), and that the difference was unlikely to be due to chance (one sample t-test against $P = 0.34$: $t(2038) = 22.0$, $p < .0001$, $x_p = 1$). Notably, this formulation entails a strongly diminishing marginal utility of R for values below 1 (1 = no discounting). For example, if an individual with $d = 0.8$ receives 100 likes, the effective utility is ~ 40 , while for 1000 likes, the utility is ~ 250 . The mean d was reliably below 1 ($M = 0.9$, $t(2038) = 39.4$), and with a skewed distribution (25th quantile = 0.006, 50th = 0.77, 75th = 1.35, 100th = 6.9), meaning that for most individuals, the marginal utility of likes was strongly diminishing. Adding the nonlinear utility function $u(R) = R^d$ to the basic $\bar{R}L$ – model better accounted for the data (AIC_W : 0.59 vs 0.41, $x_p = 1$).

The relative improvement in model fit was positively associated with Instagram follower number (linear regression of log follower number on the difference in AIC_W relative to No Learning between $\bar{R}L^d$ and $\bar{R}L$ – models: $\beta = 0.013$, $SE = 0.003$, $t = 4.07$, $p < .0001$. Regression of log follower number on AIC_W directly comparing $\bar{R}L^d$ and $\bar{R}L$ – models: $\beta = 0.011$, $SE = 0.004$, $t = 2.94$, $p = .003$), indicating that incorporating diminishing marginal utility of likes improved model fit particularly for individuals with many followers. This suggests that individuals with many followers might have more strongly diminishing marginal utility functions. We tested this notion directly by predicting (log-transformed) follower number from the (standardized) estimated d parameter using linear regression (estimates were very similar using negative-binomial regression, and when including all model parameters), controlling for differences in the number of posts (which otherwise might confound the relationship). We found that a lower estimated d parameter (i.e., more strongly diminishing marginal utility) was associated with a higher number of followers ($\beta = -0.19$, $SE = 0.044$, $t = -4.16$, $p < .0001$). In other words, the marginal utility of likes, which drives learning, was lower for individuals with more followers. Together, these results show that individuals with many followers might habituate to social rewards, necessitating more likes for an equivalent motivational effect.

In contrast, we found no evidence for diminishing marginal utility in Study 2, as the basic $\bar{R}L$ – model fit best in all three datasets (see Table S2). This is likely due to the, on average, much lower number of likes per post in Study 2 than Study 1 (see Table S1).

Table S2

Dataset	$u(R) = R$	$u(R) = sR^d$	$u(R) = R^d$	$u(R) = sR$
Men's Fashion	0.42	0.15	0.26	0.15
Women's Fashion	0.46	0.08	0.27	0.19
Gardening	0.37	0.12	0.25	0.26

Table S2. Comparison of utility functions in Study 2 showed, in contrast to Study 1, no evidence for diminishing marginal utility of likes in reinforcement learning. The table shows the relative evidence (AIC_w) for different utility functions added to the $\bar{R}L$ – model. $u(R) = R$ denotes the standard $\bar{R}L$ – model. The exceedance probability (xp) for $u(R) = R$ was 1 in all Study 2 datasets.”

The authors do consider and attempt to test one interesting possibility: that people may compare the rewards they receive to those of others. However, I found the way that social comparison was operationalized to be somewhat strange. Although it was somewhat difficult to evaluate based on lack of detailed information about the data, it seems like the social comparison models added a term to the prediction error that incorporates the average number of likes per post on the forum in the preceding week. So, the “likes” for a given post get scaled by the average number of likes for all posts on the forum in the past week. In concrete terms, if I post on the forum and get 10 likes, this carries less impact if in the past week, posts on the forum on average got 100 likes, compared with a situation where posts in the past week on average got 5 likes. I question whether this operationalization can accurately capture social comparison effects at the individual user level. This is because, it seems like it would be impossible to tell from the available data, how aware an individual user is of the average number of likes per post on the forum. Let's say I post on October 1st, I get 5 likes, and then I post a second time on October 10th, and get 10 likes. From October 3rd – 10th, on average posts in the forum are getting 50 likes. Lindstrom's social comparison model would code my experience as a negative prediction error, since the number of likes I got was lower than the forum average over the past week. But, I haven't logged in since the 1st. So how would I know that the forum average is 50 likes between the 3rd and the 10th? I am not sure how the authors would be able to

determine individual users' knowledge of whether they are getting more or fewer likes than others in their network. But it seems like it would be necessary to verify this knowledge somehow, in order to be able to make claims about social comparison. To be sure, I do not doubt that social comparison on social media is a real phenomenon. I am just not sure the authors are capturing it in a meaningful way in their data. They also do not provide evidence the social comparison model explains the group level data better than the basic RL model. This may be because they are not measuring social comparison appropriately.

Response: The reviewer raises an interesting point. It is correct that we do not have any direct measure of an individual user's knowledge of, or attention to, the likes others receive. In other words, our inference about social comparison is statistical and probabilistic in nature – we test whether the data is consistent with social comparison, as specified in the model. Note that a very similar logic underlies inferences about processes in many behavioral lab studies - the effects of experimental manipulations (e.g., an attentional manipulation) are often inferred via statistical tests of the outcome (e.g., reaction times) rather than direct measurement of the information acquisition (e.g., eye tracking). We agree that it might be useful for future studies to have detailed information about what people attend to, but this is unfortunately not feasible given the current data. Still, if a user only logs in to post, but never observes the likes other users receive, the social comparison model would give a poor account of this user's data (and thereby not be favored in model comparison), as the average likes others receive has no way to affect the users behavior.

Note that a very similar logic underlies inferences about process in many behavioral lab studies - the effect of experimental manipulations (e.g., an attentional manipulation) are often inferred via statistical tests of the outcome (e.g., reaction times) rather than direct measurement of the information acquisition (e.g., with eye tracking). We agree that it would be idea to have detailed information about what people attend to, but this is unfortunately not feasible given the current data.

In addition, we bolster the evidence for social comparison. We have improved the fit of the social comparison model by simply using the median, instead of mean, likes in the preceding week as input for social comparison. In retrospect, it's obvious that the median gives a better estimate (i.e., has higher likelihood) of the density of a long-tailed, skewed distribution (likes followed a long tailed Poisson distribution). In other words, the median number of likes gives a better estimate of what's typical on the forum than mean likes.

With this improved modeling, we find that the social comparison model indeed fits best on average. However, because we have added an experiment, our revised manuscript now puts less emphasis on the social comparison results, moving much of it to the Supplementary Material. We hope the reviewer agrees that these revised results indeed show that social comparison matters.

We report the revised model comparison results in the main text (pages 18-19):

“The preceding analyses showed that people dynamically adjust their social media behavior in response to their own social rewards, as predicted by reward learning theory—a theory originally developed to test nonsocial rewards (e.g., food reward) in solitary contexts. However, given the intrinsically social context of social media use, we speculated that reward learning online could be modulated by the rewards others receive. Thus, we next asked whether individual differences in social comparison (Rosenthal-von der Pütten et al., 2019) might account for additional variation in how reward learning mechanisms guide social media behavior. Because the format of the type of social media sites analyzed in Study 2 facilitates direct social comparison (one’s post, and the likes it incurred, are displayed in sequential order together with others’ posts on the same forum and topic), we focused our analysis on these datasets. As a model-based test for social comparison in reward learning, we modified the $\bar{R}L$ model to include an additional term, $-\xi \bar{R}_{Social}(t)$. Here, $\bar{R}_{Social}(t)$ refers to the median number of likes per post on the forum (within the threads we analyzed, see SM) in the week preceding t (we compared different time windows, see SM, section “*Additional analysis of social comparison model*”), and ξ to a free parameter that determines the strength of social comparison. The social comparison term functions as a time-specific social reference level, which in practice can transform even objectively large rewards into negative prediction errors if others receive even larger rewards. As these datasets lack information regarding the specific social information to which individual users attended, our test of social comparison is by necessity probabilistic. In other words, the model comparison tests how well the data adhered to patterns expected under a specific definition of social comparison. This approach allowed us to quantitatively test the likelihood that social comparison contributes in any degree to reward learning dynamics on social media.

Our data suggested that social comparison matters: the $\xi + \bar{R}L$ model explained posting dynamics better than the $\bar{R}L$ model (and the No Learning model) for the majority of users (mean $AIC_w = 0.46$, paired t-test of AIC_w for $\xi + \bar{R}L$ model vs $AIC_w \bar{R}L$ model: $t(2128) = 3.89$, $p = .0001$, see Figure S4A), although this group-level evidence was relatively weak. In contrast, fixed effects model comparison strongly favored the $\xi + \bar{R}L$ model, which indicates that social comparison played an especially important role for a subset of individuals (see SM, section “*Additional analysis of social comparison model*”). Notably, social comparison in the $\xi + \bar{R}L$ model only occurs upwards (i.e., reflecting disadvantageous inequality or envy (Fehr & Schmidt, 1999)): the rewards one receives become less valuable if others receive more (Fliessbach et al., 2007). Models that also included downward social comparison

(advantageous inequality or pride/gloating(Wills, 1981)) provided an inferior account of the data—a pattern that further adheres to known dynamics of social comparison²⁰ (see SM and Table S3). These results suggest that social comparison contributes to reward learning on social media.”

And in the Supplementary Material (section “*Additional analysis of social comparison model*”):

Fixed effects model comparison of the social comparison model in Study 2. Fixed effects model comparison assumes that all individuals use the same model, which entails by summing the raw AIC scores across individuals. This procedure is sensitive to outliers, individuals for whom a model fits especially well/poorly. As such, the methodology is informative for determining if a model provides a disproportionate benefit for a subset of individuals. As outlined in the main text, the fixed effects model comparison results (Table S9) indicates that social comparison played an especially important role for some individuals.

Table S9

Dataset	$\xi + \bar{R}L$ model	$\bar{R}L$ model	ΔAIC
Study 2: Men’s Fashion	216368.9	222034.8	-5667.95
Study 2: Women’s Fashion	195398.3	197399.4	-2001.06
Study 2: Gardening	546176.3	553383.3	-7197

Table S9. Fixed effects model comparison of the $\xi + \bar{R}L$ model and the $\bar{R}L$ model. The table shows the summed AIC for each model. ΔAIC denotes the difference between the $\xi + \bar{R}L$ model and the $\bar{R}L$ model, where negative values indicates evidence for the $\xi + \bar{R}L$ model.

Difference in social comparison between social network sites. The strength of the social comparison, captured by ξ , was highly variable across individuals (Figure S4B) and differed among the three social network sites. Specifically, median ξ was higher on the Women’s Fashion site than on both the Men’s Fashion (Brown-Mood median test, $z = 3.64$ $p = .0003$) and Gardening sites ($z = 4.27$, $p < .0001$), while there was no reliable difference among the latter two ($z = 1.4$, $p = .17$).

Figure S4. (A) Model comparison showed that the majority of individuals in Study 2 was best described by the $\xi + \bar{R}L$ model, which includes social comparison. Bars show mean AIC_W . Error bars denote 99% CI. Red crosses indicates exceedance probability for each model. **(B) Distribution of the estimated ξ parameter.** The figure includes the subset of individuals for whom the $\xi + \bar{R}L$ model had the highest AIC_W on the three social media sites. The vertical lines indicate the median estimated ξ parameter for each social network site.”

A second concern arises with a key assumption of the authors' favored model. This model includes an effort cost C that penalizes posting in quick succession. This apparently is imported from animal models of reinforcement learning, where obtaining rewards typically requires physical effort. While the assumption that C is positive makes sense in the context of animal models where effortful physical actions are involved, I am not sure it makes sense in the context of social media. In the supplemental materials, the authors write “posting three times in one day is more costly than posting three times in three days”. This seems like another area where RL in the social media context could be quite different from traditional RL contexts. Logging into social media and “catching up” on the current conversation – particularly in a discussion forum – has a fixed cost (or maybe even a cost that scales positively with time, if it takes longer to catch up the longer you have been offline). Thus, posting three times in quick succession within the same login session would be expected to actually be less costly than posting three times in three separate login sessions. There may be some nonlinearities in the effort cost function, whereby posting several times in quick succession within the same login is the least costly, and then following that the effort cost is flat. From what I can tell, the authors never tested the possibility that effort costs are negative at very short timescales and/or increasing positively, rather than negatively, with time. More generally, the entire social media

industry is designed to make posting as effortless as possible, so this seems like a real opportunity to test precisely how effort costs might differ when RL is put in the social media context.

Response: The reviewer makes a valid and interesting point. As the reviewer notes, the effort cost formulation, in which the effort cost is decreasing with time between responses, comes from models of animal behavior. Nonetheless, we think there are good reasons as to why the formulation also makes sense in the context of social media, as the effort costs associated with social media use (and other digital communication forms such as email) on modern devices are typically not due to logging in, but *performance* related.

To illustrate, if one has the Instagram (or any social media) app installed on the phone, the cost of logging in is negligible (one click). While the cost of “catching up” (going through all recent posts) might increase with time, this is not the process we are modeling; what we modeled is the act of posting. Posting on the social media platforms we analyzed typically involves taking pictures, selecting the right one, editing it, adding filters, tagging it, and adding any text. Together, this arguably carries significant performance-related effort cost. Moreover, if one has limited time or energy for posting, the performance-related effort cost of posting will decrease with time elapsed between posts. To give an alternative illustration, if one has access to email (e.g., via the Gmail app) on a smartphone, logging in has a negligible cost (one click), while the act of *writing* emails is arguably associated with a performance-related effort cost (which varies with content and importance of the email). In our model, the subjective effort cost of writing an email would increase when one writes several emails in close temporal proximity (which might especially be the case if every written email is important). In other words, one might feel like taking a break (as the energy runs out) before writing the next email. We believe that this is equivalent to the concept of motivational fatigue (Müller & Apps, 2019), which provides an important principle in the tradeoff between effort and reward in many decision-making situations.

Nonetheless, we have evaluated the reviewer’s suggestions of fixed and positive effort cost terms by conducting additional model comparisons. To summarize, we do not find any evidence that these alternative effort cost formulations improve the explanation of the data. Rather, model comparison supported the original effort cost formulation, but also suggests that the effort cost term does not necessarily have a strong impact on model fit, especially if response latencies are short. We summarize the results in the Supplementary Material (section “Model comparison with alternative models”):

“The effort cost term of the $\bar{R}L$ model (main text, eq. 2) is an exponentially decreasing function of τ_{Post} . This formulation is based on established RL theory for free operant tasks

(Niv et al., 2007). However, it is possible that the effort cost on social media takes different forms. We evaluated two alternative effort cost formulations: (i) exponentially increasing with time, and (ii) fixed. We did not find any evidence that either a fixed effort cost, or a cost that increased with post latency improved model fit (Table S4).

For the Instagram dataset, the fit of the original $\bar{R}L$ model and the model with increasing effort cost was similar (direct comparison: $AIC_w = 0.502$ vs 0.497 , $x_p = .71$). This might be because most users in the Instagram dataset posted with on average short latencies, which would render the influence of a time dependent (either positive or negative) effort cost term less pronounced. In line with this reasoning, we find that the original $\bar{R}L$ model, with negative effort cost, fits relatively better for users with relatively longer average posting latencies, for whom the difference between the two effort cost variants would be most impactful on model fit (Spearman $\rho = 0.12$, $p < .0001$). In the three other datasets, where average posting latencies were longer (see Table S1), we find that the original $\bar{R}L$ model, where effort cost decreases with τ_{posts} , best explained the data (see Table S4 below). We also find that the original decreasing effort cost formulation provides the overall best fit when the four datasets are pooled (combined $AIC_w = 0.46$, t-test against equal weights: $(t(4165) = 20.57, p < .0001, x_p = 1)$). Together, these results support the theory-based effort cost formulation of the $\bar{R}L$ model, but suggest that the effort cost term is not critical for model fit, especially if the average τ_{post} is short.

Table S4

Dataset	$\bar{R}L$ model	Fixed cost	Increasing cost
Study 1: Instagram	0.40/0.68	0.20/0	0.40/0.32
Study 2: Men's Fashion	0.41/1	0.32/0	0.27/0
Study 2: Women's Fashion	0.61/1	0.22/0	0.17/0
Study 2: Gardening	0.49/1	0.36/0	0.16/0

Table S4. Comparison of the $\bar{R}L$ model to alternative effort cost formulations. See text for details. The table shows the mean AIC_w /exceedance probability for each model and dataset.”

And reference these results in the main text (page 15):

“Alternative models. To test the specificity of the $\bar{R}L$ model, we compared it with a set of plausible alternative models in the SM. Specifically, we examined models (i) without effort cost (C) or net reward rate (\bar{R}) parameters, (ii) where the effort cost was fixed, or increased (rather than decreased) with post latencies, (iii) without an instrumental response policy, and (iv) based on foraging theory. Each of these provided worse accounts of the data than the $\bar{R}L$ model (see SM, section “*Model comparison with alternative models*”).”

A third question stems from the authors’ decision to limit their analysis of the forum data to posts containing images only. What percentage of all posts do these comprise, i.e., how much of the data was excluded on the basis of this decision? Do the results replicate if the authors consider all posts, not just posts containing images?

Response: In our main analysis of Study 2, we only included posts with user-generated images, in order to be consistent with Study 1 (Instagram, where all posts are image based), and exclude directly communicative interactions (e.g., written responses to comments directly aimed towards oneself). Image posts comprised 36% of the posts from the Men’s Fashion dataset, 28% of posts from the Women’s fashion dataset, and 20 % of posts from the Gardening dataset. When repeating our key model comparison to the datasets including all posts, we find that the $\bar{R}L$ model fits the data better. In fact, the relative evidence was even stronger than in our original analysis, likely due to the larger datasets. Critically, this analysis suggests that our findings are not limited to only image-based posts. We summarize these results in the Supplementary Material (section “*Results in Study 2 holds when including non-image posts*”):

“In our main analysis of Study 2, we only included posts with user-generated images, in order to be consistent with Study 1 (Instagram, where all posts are image based), and exclude directly communicative interactions (e.g., responding in writing to a comment directly aimed toward oneself). Image posts comprised 36% of the posts from the Men’s Fashion dataset, 28% of posts from the Women’s fashion dataset, and 20 % of posts from the Gardening dataset.

To assess the effect of this decision, we repeated our primary model-based analysis but included all posts acquired for users with at least 10 image-based posts (i.e., the same users as in the main analysis). We find that the results hold when including all posts. The $\bar{R}L$ model fit the data better than the No Learning model in all three datasets: Men’s Fashion $AIC_w = 0.89$ (one sample t-test against equal model fits: $t(542) = 36.78$, $p < .0001$), Women’s

Fashion $AIC_w = 0.88$ ($t(772) = 40.62$, $p < .0001$), Gardening $AIC_w = 0.91$ ($t(812) = 46.92$, $p < .0001$. All $x_p = 1$). ”

We reference the results from the main text (page 16):

“[We] verify in the SM (section “*Results in Study 2 holds when including non-image posts*”) that the results are qualitatively identical when including text-based posts”.

Two further questions about timescales that might be theoretically interesting. First, have the authors considered the possibility that “likes” might become less subjectively valuable the less frequently users are posting? Much has been written (albeit not in an academic context) about the false appeal of social media rewards, and how “digital detox” experiences make users realize that social feedback on social media is more hollow than offline experiences. This could be another way that RL differs online vs offline. Relatedly, the authors mention they are examining RL at “timescales rarely or never investigated in the laboratory”. It would be great to see a more detailed analysis of what timescales RL is occurring at on social media.

Response: We appreciate the reviewer’s interesting suggestion. In a sense, decreasing value for more infrequent posts is already captured by the original $\bar{R}L$ model, but for a different reason: the opportunity cost term reduces the utility of likes in proportion to the time between posts, because the opportunity cost is a linear function of time.

To more directly test the reviewer’s suggestion that likes (i.e., R) become less valuable the less frequently the user is posting, we implemented a version of the $\bar{R}L$ model where R was directly scaled with the post latency ($Utility(t) = R(t)/\tau_{post}(t)$). The model was otherwise identical to the original $\bar{R}L$ model. Note that the effect of this term is mathematically very similar to the alternative effort cost model (increasing effort cost with post latency) that we evaluate above: the utility of an outcome decreases with τ_{post} . As for that comparison, we do not find any evidence that the reward value (utility) of likes decreased with posting latency in the data (see Table A below).

Similar to the effort cost comparison, we find that for the Instagram dataset, the fit of the original $\bar{R}L$ model and the modified “Decreasing R ” model was almost equivalent. This might again be because the average response latency on Instagram was short (as shown in the new Table S1), which might render the influence of the time variable R term less notable. In line with this reasoning, we find that the evidence for the original $\bar{R}L$ model is stronger for users with relatively longer average posting latencies, for whom the difference between the two models (original $\bar{R}L$ model and “Decreasing R ”

model, see Table A below) would be most impactful on model fit (Spearman $\rho = 0.12$, $p < .0001$). Nonetheless, the combined evidence that the original $\bar{R}L$ model fits the data best is clear (pooled $AIC_W = 0.61$, t-test against equal model weights, $t(4167) = 18.9$, $p < .0001$, exceedance probability = 1).

We hope that the reviewer agrees that our new analyses clarifies that the assumptions underlying the original $\bar{R}L$ model were warranted. We have opted not to include these results in the revised manuscript, because, as outlined above, the $\bar{R}L$ model already incorporates decreasing utility for longer delays.

Dataset	$\bar{R}L$ model	Decreasing R
Study 1: Instagram	0.50	0.50
Study 2: Men’s Fashion	0.53	0.47
Study 2: Women’s Fashion	0.69	0.31
Study 2: Gardening	0.62	0.38

Table A. Comparison of the $\bar{R}L$ model to a model where the utility of R decreased with post latency ($R(t)/\tau_{Post}(t)$). See text for details. The table shows the mean AIC_W for each model.

The computational phenotyping was interesting, but difficult to interpret. Do the authors have any theory-driven hypotheses that could guide their analysis of individual differences?

Response: The purpose of the computational phenotyping analysis was to determine if there are distinct “types” of reinforcement learners on social media in a data-driven (unsupervised) manner, which was made possible by our large dataset ($n > 4000$). We believe that mapping such variability among users is important given the high prevalence of social media use, and the limited previous knowledge about reward learning processes on social media. Using computational phenotyping for this mapping allows directly relating different patterns of reward learning (i.e., computational phenotypes) on social media to other learning and decision-making processes (e.g., decisions weighing efforts costs and benefits), which would not be possible without a model-based analyses.

However, future work is needed for relating the different phenotypes to external measures, both pertaining to social media per se (e.g., social media addiction), and to psychological traits (e.g., personality). Furthermore, our results might, for example, serve as basis for developing questionnaires

about social media use (where one might expect 4 factors corresponding to the 4 phenotypes), and for inspiring different types of “interventions” for problematic social media use (e.g., where costs and rewards are differentially manipulated for different phenotypes).

In summary, the computational phenotyping analysis was data-driven and exploratory rather than hypothesis testing. We have clarified this in the revised manuscript (page 20):

“More specifically, we used the three parameters of the original $\bar{R}L$ model (which is nested in the model variations estimated for Study 1 & 2), estimated for each individual from datasets 1-4 (total $n = 4,168$), as input for k-means clustering, an unsupervised, data-driven method for finding sub-groups in multidimensional data.”

Finally, I found it difficult to evaluate some of the methods and results due to missing information. The authors should report descriptive statistics for all key variables, e.g. to get a sense of what “high” and “low” reward rates mean, in concrete terms. How many likes on average do posts get? How frequently does the average user post?

Response: We agree with the reviewer that this information is beneficial, and apologize for omitting it in the original submission. We have added these descriptive statistics to Table S1 in the supplementary material:

” Table S1

Dataset	Likes				τ_{Post}				Followers	
	M Pop.	Md Pop.	M Ind.	Md Ind.	M Pop.	Md Pop.	M Ind.	Md Ind.	M Pop.	Md Pop.
Study 1: Instagram	377	43	338	74	1.82	0.95	3.12	1.45	324	145
Study 2: Men’s Fashion	17.46	14	16.46	12	14.61	2.99	33.64	6.99	-	-
Study 2: Women’s Fashion	16.45	16	17.22	16.5	11.96	1.75	24.47	4.55	-	-
Study 2:	6.1	4	5.43	4	8.03	1.01	22.2	2.98	-	-

Table S1. Descriptive statistics for likes, τ_{Post} , and followers across the four datasets. The table shows both sample averages (Pop.), and individual-level averages (Ind.). The latter was generated by computing the mean (M) and the median (Md) for each individual, and then summarized with mean-of-means and median-of-medians of the individual level averages.”

How does that frequency change with every 10% increase in reward rates?

Response: To address this question, we reparametrized the regression models by scaling the subjective reward rate to 1-100 for each user. We have added the estimates (for each 1% change in the reward rate) to the revised manuscript (pages 14 and 17):

“Thus, increasing the average subjective reward rate from low to high reduced average latency between posts by ~18%, corresponding to ~8 hours. Based on analysis with a continuous \bar{R} term, this corresponds to a reduction of 0.34% (~5 minutes) in average posting latencies for each 1% increase in the subjective reward rate.”

And

“Thus, in these respective datasets, latencies between posts were 8%, 16%, and 18% shorter when the average reward rate was high rather than low. This corresponds, based on analysis with a continuous \bar{R} term, to a reduction of 0.18%, 0.41%, and 0.38% in average posting latencies for each 1% increase in the subjective reward rate.”

It would also be helpful to see the key models written out in the main text rather than the supplement.

Response: Following the reviewer’s suggestion, we have moved the key model description from the SM to the Methods section of the main text.

Other comments:

-Fig 2C y-axis label is missing

Response: We originally opted for no y-axis label, because multiple variables are displayed in the figure. In the revised manuscript, we label the axis as “x” to denote this.

-I didn't have a good sense of how good the model fits were in general. How do they compare, for example, to model fits of RL behavior in lab settings?

Response: To provide an intuition for the quality of the model fits, we conducted Spearman correlations between the data and model prediction (i.e., the policy for the social media data). A correlation of 1 would indicate a perfect monotonic relationship between the model predictions and the data. Note that there are no standard measures of absolute model fit that generalize between different types of data (e.g., exponentially distributed for the social media data, binomially distributed for lab choice data), meaning that the Spearman correlations should be taken as providing an intuition rather than an exact measure. We both conducted correlations across the entire data set (pooling users), and on the individual level, using mixed-models (with rank transformed data and predictions), which takes differences between users into account (see Table below).

Dataset	Spearman ρ (pooled)	Spearman ρ (LMM)
Study 1: Instagram	0.58	0.48
Study 2: Men's Fashion	0.48	0.44
Study 2: Women's Fashion	0.47	0.46
Study 2: Gardening	0.46	0.39

As comparison, we conducted the same analysis for model fits to decision making in the lab (Lindström et al, *PNAS*, 2019, experiment 1 & 2). In two datasets (each $n = 40$, 70 data points per participant), we find that the correlations were of similar magnitude as in the current study (dataset 1, pooled $\rho = .57$, LMM = 0.54. dataset 2: pooled $\rho = 0.52$, LMM = 0.49). This comparison indicates that the quality of model fits to the social media data was comparable to lab data.

-Did models control for whether a given user started a thread?

Response: The models in Study 2 did not control for if a user started a thread, nor did it distinguish between different threads. We sought to keep the Study 2 analysis as close to Study 1 (Instagram) as possible, by simply ordering posts chronologically for each poster. We have clarified this in the revised supplementary material (section “*Additional information about Study 2*”):

“For simplicity and consistency with Study 1, the model-based analysis did not distinguish between threads.”

In the statistical analysis of the effect of the reward rate on posting latencies, we accounted for mean differences in posting response latencies between threads by additional dummy predictors, as described in the SM (section “*Additional information about Study 2*”):

“To adjust for potential differences between threads in average posting latency, the statistical analyses included either fixed (Men’s and Women’s fashion forum) or random (Gardening forum) effects for thread.”

-SOM p 6, please provide more information about the “linear model” against which the QLE model was compared.

Response: The linear model was a basic linear regression with two parameters, an intercept and a slope. We have clarified this in the revised Supplementary Material (section “*Estimation of the quantitative law of effect*”):

“The linear model consisted of a linear regression of response rates on reward rates. Both the QLE (k and R_0) and the linear model (intercept and slope) had two free parameters.”

-SOM p 8, reports that outliers falling outside 20th-80th percentiles were removed, but main text reports 10th-90th. Which is correct?

Response: The correct is 20th-80th percentiles. We thank the reviewer for pointing out this inconsistency, which has been corrected in the revised version.

-Fig S2, please include shading of model predictions as in Fig 2C

Response: We have followed the reviewers suggestion, and added shaded model predictions:

Figure S2. Additional example individuals in Study 1 (Instagram). To supplement Figure 2C (main text), we here show nine (out of 2,039) additional randomly selected example individuals. Purple lines indicate τ_{post} , yellow lines \bar{R} , and red lines the model policy/threshold. The faded purple lines show 100 simulations of τ_{post} from the estimated model policy, which illustrate the expected degree of variability given that policy, and how the empirical τ_{post} typically falls within this range. The x-axis shows the number of social media posts per individual. These examples show the wide variability in social media posting patterns.

REVIEWER COMMENTS

Reviewer #1 (Remarks to the Author):

The authors have provided an extensive revision of their manuscript and I am particularly impressed that they were able to add in the results of an experimental manipulation. Thank you for addressing my comments.

Reviewer #2 (Remarks to the Author):

The authors have done an impressive job in handling the constructive comments.

Reviewer #3 (Remarks to the Author):

In their revised paper, Lindstrom et al present additional analyses and data to provide support for their RL account of social media engagement, including a new experiment. Most of my original comments have been addressed and I appreciate the authors' efforts in this revision, especially the provision of new experimental data which significantly strengthens their claims about the causal effects of reward rate on posting latencies, and the consideration of alternative models. The authors have clearly put in a lot of additional work and I enjoyed reading the revision.

One of the most novel and interesting claims in the paper is that "social comparison contributes to social reward learning." This is perhaps the most substantial theoretical advance this work can provide, since it captures a phenomenon that may distinguish online RL from RL typically studied in the lab.

However, I continue to have concerns, raised in my original review, about whether the evidence provided by the authors supports their claims about social comparison. Social comparison is operationalized with a parameter that captures sensitivity to the average number of likes per post on the forum, such that in individuals with a high sensitivity, effects of own rewards are down-weighted if others are receiving more rewards. The issue here is that the authors seek to make claims about a psychological process at the user level (social comparison) from a variable that is measured at the network level (average number of likes per post in the network). I suggested that in order to make claims about social comparison, they would need to verify that individual users attend to the information that the authors take as a proxy for a social comparison process (average number of likes per post).

In their response, the authors acknowledge that they do not have any evidence that individual users attend to the rewards that others receive. They argue that this is not a problem because "a very similar logic underlies inferences about processes in many behavioral lab studies - the effects of experimental manipulations (e.g., an attentional manipulation) are often inferred via statistical tests of the outcome

(e.g., reaction times) rather than direct measurement of the information acquisition (e.g., eye tracking).” They further claim that if users are not aware of the rewards others receive, the social comparison model would provide a poor fit to the data. They revised their formulation of the social comparison model by replacing mean likes per post with median likes per post and show that the social comparison model now fits best on average.

I’m not convinced by this argument. In lab studies experimenters have a high degree of control over stimulus presentation, and inferring the effects of a manipulation of attention from response times is straightforward because there is a large literature establishing a link between attention and response times, and because in a lab setting the experimenters are able to limit the possibility that other variables apart from the attentional manipulation can explain the variation in response times. In contrast, here the authors seek to make an inference about a psychological process (social comparison) in a non-experimental setting where any number of unmeasured variables could covary with the independent variable of interest (average number of likes per post) and provide alternative explanations for the observed effects. For instance, the median/mean number of likes per post could have something to do with the content posted in the network, which might detract users’ attention away from the likes that they themselves are getting. We might expect that more engaging or interesting content gets more likes per post, and this engaging content distracts users from their own rewards, producing a similar effect that the authors observe in their data that is not due to users comparing their own rewards to others’ rewards, but rather because users are distracted by engaging content posted by others and pay less attention to their own rewards. This is just one plausible alternative explanation but there may be others.

To be sure, I am sympathetic to the fact that it is very challenging to infer psychological processes from social media data where there are vast number of unmeasured variables. But without experimentally manipulating users’ observations of others’ rewards in the network, or at least verifying that users are in fact aware of the rewards others’ posts receive, it’s not possible to attribute the observed results to the psychological process of social comparison.

One straightforward way to address this issue is using the experimental framework the authors designed to test the causal effects of reward rate on posting latencies. Indeed, it appears they built this framework with social comparison in mind, as they designed the experiment such that participants could not observe the likes that other users received “to preclude social comparison”. So it should be possible to demonstrate experimentally that participants’ reward learning is sensitive to a causal manipulation of the number of likes other users receive, while controlling for the content of posts. This would provide convincing evidence for the claim that social comparison processes contribute to social reward learning.

In the revision the authors report that the ‘social comparison’ model now provides the best fit on average. So I was confused why the individual differences analysis (‘computational phenotyping’) was based on the basic RL model which provides a worse fit to the data. This was particularly surprising given the evidence that fixed effects model comparison, which is informative for identifying subsets of

individuals, strongly favored the 'social comparison' model, which suggests there may be computational phenotypes related to whatever psychological process is being captured by the 'social comparison' parameter.

Finally, the authors report that users with more followers are less sensitive to social rewards, and explain this with a diminishing marginal utility function and suggest users might habituate to 'likes'. One alternative possibility is that users with a high number of followers are more likely to have accounts that are run by multiple individuals (e.g., 'influencers' or celebrities sometimes have paid assistants who post content to their account). Can this be ruled out? If not, this is a limitation that should be mentioned.

In their revised paper, Lindstrom et al present additional analyses and data to provide support for their RL account of social media engagement, including a new experiment. Most of my original comments have been addressed and I appreciate the authors' efforts in this revision, especially the provision of new experimental data which significantly strengthens their claims about the causal effects of reward rate on posting latencies, and the consideration of alternative models. The authors have clearly put in a lot of additional work and I enjoyed reading the revision.

Response: We sincerely thank the reviewer for the positive comments about our revised manuscript.

One of the most novel and interesting claims in the paper is that “social comparison contributes to social reward learning.” This is perhaps the most substantial theoretical advance this work can provide, since it captures a phenomenon that may distinguish online RL from RL typically studied in the lab.

Response: We very much appreciate the reviewer's thoughts on this. We agree that the social comparison effect on rewards is a novel and exciting contribution of this work to RL. However, in line with the reviewer's point, we strongly believe that in order to seriously explore this idea, it will require a stand-alone set of much more focused studies.

The main aim and contribution of the current manuscript is to first establish RL as a serious explanatory mechanism for online social media behavior. This is novel, too, and we (and the other reviewers) believe that it offers a major theoretical advance in its own right. That is, it provides the first rigorous evidence for a deep similarity between modern human behavior on social media and cross-species reward learning, and it presents a new computational framework for research on social media addiction. To clarify our main contribution further, we have conducted extensive simulations of our reinforcement learning model. The purpose of these simulations is to show that the model can reproduce standard behavioral patterns of animals in Skinner boxes (section “*Computational Modeling: Simulation of key empirical regularities in operant conditioning research*” in the revised Supplementary Material). In turn, this further demonstrates the theoretical validity of our model as an account of free operant reward learning, and as a tool to identify reward learning on social media.

As a related note: Reviewer 3 suggests that social comparison “may distinguish online RL from RL typically studied in the lab,” and this is part of their argument for asking us to expand the social comparison component in our paper (with new data). However, social comparison is not unique to online RL; rather, social comparison can play a role in any context involving social reward. Thus, we view the reviewer's note as emphasizing the need for more research on social aspects of RL in general, but not as an argument for expanding our current analysis of social comparison.

We have made revisions to clarify these points. We hope that the reviewer will agree with this perspective—that our main finding is a necessary first step, exciting in its own right, and that the social comparison idea deserves a full-blown follow-up program of work. Thus, our main revisions to the present paper acknowledge the

limitations of our conclusions regarding social comparison and, in doing so, lay the groundwork for new work dedicated to this idea. Indeed, we plan to address these issues in a separate, new program of work.

However, I continue to have concerns, raised in my original review, about whether the evidence provided by the authors supports their claims about social comparison. Social comparison is operationalized with a parameter that captures sensitivity to the average number of likes per post on the forum, such that in individuals with a high sensitivity, effects of own rewards are down-weighted if others are receiving more rewards. The issue here is that the authors seek to make claims about a psychological process at the user level (social comparison) from a variable that is measured at the network level (average number of likes per post in the network). I suggested that in order to make claims about social comparison, they would need to verify that individual users attend to the information that the authors take as a proxy for a social comparison process (average number of likes per post).

In their response, the authors acknowledge that they do not have any evidence that individual users attend to the rewards that others receive. They argue that this is not a problem because “a very similar logic underlies inferences about processes in many behavioral lab studies - the effects of experimental manipulations (e.g., an attentional manipulation) are often inferred via statistical tests of the outcome (e.g., reaction times) rather than direct measurement of the information acquisition (e.g., eye tracking).” They further claim that if users are not aware of the rewards others receive, the social comparison model would provide a poor fit to the data. They revised their formulation of the social comparison model by replacing mean likes per post with median likes per post and show that the social comparison model now fits best on average.

I’m not convinced by this argument. In lab studies experimenters have a high degree of control over stimulus presentation, and inferring the effects of a manipulation of attention from response times is straightforward because there is a large literature establishing a link between attention and response times, and because in a lab setting the experimenters are able to limit the possibility that other variables apart from the attentional manipulation can explain the variation in response times. In contrast, here the authors seek to make an inference about a psychological process (social comparison) in a non-experimental setting where any number of unmeasured variables could covary with the independent variable of interest (average number of likes per post) and provide alternative explanations for the observed effects.

For instance, the median/mean number of likes per post could have something to do with the content posted in the network, which might detract users’ attention away from the likes that they themselves are getting. We might expect that more engaging or interesting content gets more likes per post, and this engaging content distracts users from their own rewards, producing a similar effect that the authors observe in their data that is not due to users comparing their own rewards to others’ rewards, but rather because users are distracted by engaging content posted by others and pay less attention to their own rewards. This is just one plausible alternative explanation but there may be others.

Response: We agree with the reviewer that we cannot rule out all possible explanations for our observation that the median number of likes in the week preceding a post influences learning for some individuals, because data on individual attention simply is not available.

We have thoroughly revised the social comparison section and more clearly state that these results are preliminary and our conclusions regarding social comparison speculative for the time being.

To this end, we have (i) changed the section header (from “Social comparison contributes to reward learning” to “Social comparison in social reward learning”), and (ii) changed the sentence “This analysis suggests that social comparison matters.” to “This analysis suggests that social comparison may matter.” Furthermore, (iii) we have removed the sentence “This approach allowed us to quantitatively test the likelihood that social comparison contributes in any degree to reward learning dynamics on social media.” We also (iv) replaced the final sentence in the section (original: “These results suggest that social comparison contributes to reward learning on social media.”) with “Together, these exploratory results suggest that social comparison may contribute to reward learning dynamics on social media.”

Furthermore, we have added a paragraph to the Discussion where we clarify that the social comparison results represent preliminary findings on a novel hypothesis generated by our RL framework and opportunity for future research:

“[O]ur analysis in Study 2 suggests that social comparison may contribute to reward learning on social media by providing a social reference for the number of likes required to elicit a positive reward prediction error. Although this result comports with prior research examining social comparison on social media (Carr, Hayes, & Sumner, 2018; Rosenthal-von der Pütten et al., 2019), as well as neural reward processing (Fliessbach et al., 2007), the correlational nature of the big data used here necessitates caution, as other explanations cannot be ruled out. These findings present an opportunity for future experimental research to establish the causal nature of social comparison and to explore its boundary conditions.”

We have also added new text to highlight the fact that several studies show that people do engage in online social comparison – based on likes - in a manner consistent with our modeling and interpretations. We now briefly describe these studies, which we previously only cited in the introduction, in the section about social comparison:

“In support of this idea, previous research has shown that social comparison plays an important role in determining how many likes are required for a social media post to be experienced as successful (Carr et al., 2018), and that receiving fewer likes than close others generates negative affect (Rosenthal-von der Pütten et al., 2019).”

To be sure, I am sympathetic to the fact that it is very challenging to infer psychological processes from social media data where there are vast number of unmeasured variables. But without experimentally manipulating users' observations of others' rewards in the network, or at least verifying that users are in fact aware of the rewards others' posts receive, it's not possible to attribute the observed results to the psychological process of social comparison.

Response: The Reviewer is correct, and we agree—the issue is simply that our findings on social comparison are secondary to the primary focus of the paper.

As we describe in our reply above, previous research has demonstrated that people are aware of, and sensitive to, the likes others receive for their posts, which provides a plausible basis for social comparison. However, we definitely agree with the reviewer that we cannot rule out all other explanations of the observed result. We clarify this in the discussion:

“[O]ur analysis in Study 2 suggests that social comparison may contribute to reward learning on social media by providing a social reference for the number of likes required to elicit a positive reward prediction error. Although this result comports with prior research examining social comparison on social media (Carr, Hayes, & Sumner, 2018; Rosenthal-von der Pütten et al., 2019), as well as neural reward processing (Fliessbach et al., 2007), the correlational nature of the big data used here necessitates caution, as other explanations cannot be ruled out. These findings present an opportunity for future experimental research to establish the causal nature of social comparison and to explore its boundary conditions.”

One straightforward way to address this issue is using the experimental framework the authors designed to test the causal effects of reward rate on posting latencies. Indeed, it appears they built this framework with social comparison in mind, as they designed the experiment such that participants could not observe the likes that other users received “to preclude social comparison”. So it should be possible to demonstrate experimentally that participants' reward learning is sensitive to a causal manipulation of the number of likes other users receive, while controlling for the content of posts. This would provide convincing evidence for the claim that social comparison processes contribute to social reward learning.

Response: Again, we agree with the reviewer that an experimental approach is needed to better understand the nature of social comparison in online RL. We just feel strongly that an expansion of the social comparison section of our paper would detract from our main points. Indeed, there are already a lot of findings included in the current manuscript. We hope the reviewer will agree that these ideas merit a new, separate program of experimental work on social comparison in RL.

In the revision the authors report that the ‘social comparison’ model now provides the best fit on average. So I was confused why the individual differences analysis (‘computational phenotyping’) was based on the basic RL model which provides a worse fit to the data. This was particularly surprising given the evidence that fixed effects model comparison, which is informative for identifying subsets of individuals, strongly favored the ‘social comparison’ model, which suggests there may be computational phenotypes related to whatever psychological process is being captured by the ‘social comparison’ parameter.

Response: We based the “phenotyping” on the basic RL model for several reasons. First, because we were interested in understanding general individual difference profiles in learning on social media, it was important to analyze the data from Study 1 (which used Instagram data, from which social comparison could not be analyzed) and Study 2 (where social comparison was possible) together. The basic RL model provided a unified account of reward learning in both studies. Second, the social comparison model only involves a minor addition (the social comparison parameter) to the machinery of the basic RL model, meaning that the same basic processes are represented in both models. Third, because the social comparison results were relatively weak, and of more preliminary nature, as we outline above, using the basic model for understanding online reward learning seemed more robust.

To ascertain that this decision did not introduce bias in the phenotyping, we have conducted a new set of cluster analyses based on the social comparison model. These additional analyses demonstrate that basing the cluster analysis on the social comparison model does not change the cluster structure and produces very similar results to our original analysis.

We summarize these new analyses in the Supplementary Material:

“Social comparison does not change cluster structure. We based the phenotyping analysis on the standard $\bar{R}L$ model, rather than the $\xi + \bar{R}L$ model that included social comparison, because we used the basic model to explain the data from both Study 1 (where no analysis of social comparison was possible) and Study 2 (where analysis of social comparison was possible). To rule out that omitting social comparison might have biased cluster assignment (e.g., that individuals with a stronger tendency for social comparison should be clustered together), we performed two control analyses. First, we conducted the same cluster analysis we report in the main text, but based on the $\xi + \bar{R}L$ model (for Study 1, we used the $\bar{R}L$ model parameters, and set $\xi = 0$) to assess whether inclusion of social comparison would change the cluster structure. This was not the case: multiple quantitative criteria supported a four cluster solution. Comparing cluster assignments to our original analysis (see main text), we find a strong agreement in cluster assignment (Cramer’s $V = 0.71$, Rand index = 0.7). Furthermore, we find that the cluster centroids (the mean values of the parameters for each cluster) are extremely similar (correlation for the three overlapping model parameters: $r(10) = .99$, $p < .0001$). These results show that including social comparison does not change the cluster structure.

Second, we performed the same clustering analysis based on the $\xi + \bar{R}L$ model, but omitted the social comparison parameter ξ from the input features. This analysis controls for social comparison (as the other model parameters are adjusted for social comparison by the model estimation procedure), but does not use it for clustering. Again, we found that four clusters provided the best cluster solution. As for the preceding analysis, the cluster similarity relative to the clustering based on the standard $\bar{R}L$ model was high (Cramer V = 0.8, Rand index = 0.8), and centroids highly correlated ($r(10) = .99$, $p < .0001$). Together, these results show that basing the computational phenotyping on the $\bar{R}L$ model did not bias the results.”

We reference these results in the main text:

“[W]e used the three parameters of the original $\bar{R}L$ model (See SM, section “*Social comparison does not change cluster structure*” for control analyses based on the social comparison model)” (page 20)

Finally, the authors report that users with more followers are less sensitive to social rewards, and explain this with a diminishing marginal utility function and suggest users might habituate to ‘likes’. One alternative possibility is that users with a high number of followers are more likely to have accounts that are run by multiple individuals (e.g., ‘influencers’ or celebrities sometimes have paid assistants who post content to their account). Can this be ruled out? If not, this is a limitation that should be mentioned.

Response: Because the Instagram users were anonymized in the dataset we analyzed, it’s true that we cannot rule out such alternative accounts.

However, indirect support for our diminishing marginal utility interpretation (which is a standard concept in behavioral economics) comes from our experiment, where the relationship between self-reported Instagram follower number and sensitivity to social reward mirrored the pattern we observed in the Instagram data.

We have added a paragraph mentioning this limitation in the Supplementary Material (section: “Having more followers on Instagram is associated with diminishing marginal utility of likes”):

“It should be noted that because the Instagram data was anonymized, we cannot rule out that individuals with many followers are qualitatively different in some unobserved way (e.g., represent businesses rather than private persons). However, the diminished marginal utility interpretation is in agreement with our experimental results, where participants with more Instagram followers were less strongly affected by social rewards (see “*Additional analysis of experiment*” below).”

****REVIEWERS' COMMENTS:**

Reviewer #3 (Remarks to the Author):

Lindstrom et al have responded to my remaining concerns about their paper. I agree that the topic of social comparison & RL on social media warrants a stand-alone set of more focused studies. I am also convinced that the bulk of the paper - on basic RL mechanisms on social media - is an important advance on its own.

I do still have reservations about the claims being made about social comparison in the paper as written, and my original concerns about ruling out alternative mechanisms still hold. Readers may incorrectly infer that the question of social comparison has been definitively addressed (e.g. the paper abstract still claims that social comparison explains individual differences in social learning), when this still remains an open question for future work. Given that the authors wish to focus the paper on basic RL processes, I'd recommend moving the social comparison analyses to the supplemental materials, ensure they are clearly labeled as preliminary, and remove their mention from the abstract. This will ensure that the claims made in the main paper are clearly supported by the data. I would support publication if these minor revisions are made.

****REVIEWERS' COMMENTS:**

Reviewer #3 (Remarks to the Author):

Lindstrom et al have responded to my remaining concerns about their paper. I agree that the topic of social comparison & RL on social media warrants a stand-alone set of more focused studies. I am also convinced that the bulk of the paper - on basic RL mechanisms on social media - is an important advance on its own.

Response: We are delighted that the reviewer agrees that our findings provide an important advance, and sincerely thank the reviewer for helping us improve the paper throughout the review process.

I do still have reservations about the claims being made about social comparison in the paper as written, and my original concerns about ruling out alternative mechanisms still hold. Readers may incorrectly infer that the question of social comparison has been definitively addressed (e.g. the paper abstract still claims that social comparison explains individual differences in social learning), when this still remains an open question for future work. Given that the authors wish to focus the paper on basic RL processes, I'd recommend moving the social comparison analyses to the supplemental materials, ensure they are clearly labeled as preliminary, and remove their mention from the abstract. This will ensure that the claims made in the main paper are clearly supported by the data. I would support publication if these minor revisions are made.

Response: We have followed the reviewers advice. We have removed the social comparison results from the abstract and the introduction. We have moved the social comparison results to the Supplementary Information, where we clearly label them as preliminary. Furthermore, we clearly describe the social comparison results as preliminary in the main text (page 18):

“Study 2: Social comparison in social reward learning

The preceding analyses showed that people dynamically adjust their social media behavior in response to their own social rewards, as predicted by reward learning theory—a theory originally developed to test the effects of nonsocial rewards (e.g., food) in solitary contexts. However, given the intrinsically social context of social media use, we speculated that reward learning online could be modulated by the rewards others receive. In the Supplementary Note 9 we provide preliminary support for the hypothesis that reward learning, at least on the social media platforms we analyzed in Study 2, may be modulated by social comparison.”